# Multimodal Video Generation Models with Audio: Present and Future

## Abstract

Video generation models have advanced rapidly and are now widely used across entertainment, advertising, filmmaking, and robotics applications such as world modeling and simulation. However, visual content alone is often insufficient for realistic and engaging media experiences—audio is also a key component of immersion and semantic coherence. As AI-generated videos become increasingly prevalent in everyday content, demand has grown for systems that can generate synchronized sound alongside visuals. This trend has driven rising interest in **multimodal video generation**, which jointly models video and audio to produce more complete, coherent, and appealing outputs. Since late 2025, a wave of multimodal video generation models has emerged, with releases including Veo 3.1, Sora 2, Kling 2.6, Wan 2.6, OVI, and LTX 2. As multimodal generation technology advances, its impact expands across both daily consumer and industrial domains—revolutionizing daily entertainment while enabling more sophisticated world simulation for training embodied AI systems. In this paper, we provide a comprehensive overview of the multimodal video generation model literature covering the major topics: evolution and common architectures of multimodal video generation models; common post-training methods and evaluation; applications and active research areas of video generation; limitations and challenges of multimodal video generation.

## 1 Introduction

Video generation has advanced rapidly in recent years, with models now capable of producing realistic, high-definition outputs with strong temporal consistency and visual aesthetics Li et al. (2025a); Hu et al. (2025); Wan et al. (2025); Yang et al. (2024). However, a critical dimension of human perception has remained largely absent from previous generation of generated videos: ***sound***. In real-world experience, visual and auditory sensory information are deeply interdependent from footsteps accompany walking, the ripple of tides against the shore, and dialogue synchronizes with lip movement McGurk and MacDonald (1976); Jahncke et al. (2015). The absence of contextually appropriate audio diminishes immersion, causing videos to feel incomplete and less natural to viewers. This principle is well illustrated by the history of cinema itself: the earliest films were entirely silent, yet the industry rapidly evolved to incorporate synchronized sound, high-fidelity audio, and eventually immersive theater acoustics, each advancement driven by the fundamental recognition that visual storytelling is inseparable from its auditory counterpart Sergi (2013); Babbar (2024); Goncalves et al. (2024a). Multimodal video generation, the joint synthesis of video and semantically aligned audio, has recently emerged as a distinct and rapidly growing research direction, addressing this gap with video-audio fusion architectures and new available training data. Unlike traditional video generation, which only generates visual outputs Yang et al. (2024); Zheng et al. (2024); Peng et al. (2025), multimodal video generation must solve fundamentally different challenges: *cross-modal temporal alignment* Haji-Ali et al. (2025); Wang et al. (2024), *semantic coherence between audio and visual streams* Ruan et al. (2023a); Luo et al. (2023a), and *the generation of plausible soundscapes that adapt to scene dynamics* Lee et al. (2025); Chen et al. (2025a). This makes multimodal video generation not merely an extension of video synthesis, but a qualitatively different problem requiring different architecture designs and training paradigms. The release of several state-of-the-art models in late 2025 and early 2026, including Sora 2 Liu et al. (2024a);

OpenAI (2025a), Veo 3.1 Wiedemer et al. (2025), Grok 4 xAI (2025), Wan 2.6 Wan et al. (2025), Kling 2.6 Kuaishou Technology (2025), OVI Low et al. (2025), and LTX 2 HaCohen et al. (2026), has revealed this paradigm shift: video generation is increasingly expected to be multimodal by default. This trend is driven by growing demand across diverse application and active research domains, from advertisement production Hu et al. (2025); Anantrasirichai et al. (2026) and short film creation Zhang et al. (2025a); Huang et al. (2025a); Leininger et al. (2025) to audio-visual video editing Team et al. (2025); Guo et al. (2025); Liu et al. (2025a); Ishii et al. (2025) and social media entertainment Anderson and Niu (2025); Ye et al. (2025). Designers, researchers, and industry practitioners alike are actively exploring both the capabilities and the underlying architectures of these systems Ma et al. (2025a). While several existing surveys comprehensively review video generation Wang et al. (2025a); Ma et al. (2026); Hayawi and Shahriar (2025); Elmoghany et al. (2025); Bhagwatkar et al. (2020), they predominantly focus on the visual modality alone. This paper specifically addresses multimodal video generation with sound.

We provide a systematic review of the most recent multimodal video generation that includes: 1. foundations and evolution of architectures of modern multimodal video generation; 2. common post-training methods and evaluation strategies 3. applications and active research areas; 4. limitations of current multimodel video generation models . Our literature review aims to provide the most up-to-date overview of multimodal video generation.

## 2 Components of Multimodal Video Generation Architectures

Unlike traditional visual-only generation, multimodal video generation aims to model the joint distribution of visual frames and audio waveforms HaCohen et al. (2026); Cheng et al. (2025a); Ruan et al. (2023b). The core challenge lies in synchronizing these distinct modalities within a unified architecture. In this section, we trace the architectural components of existing open-source models, specifically highlighting how they integrate audio synthesis with video dynamics. A summary of widely-adopted multimodal video diffusion models is presented in Table 1.

### 2.1 Variational Autoencoder (VAE)

While the Variational Autoencoder (VAE) Kingma and Welling (2013) established a foundational architecture for probabilistic generative modeling, its primary usage has shifted. Originally it served as a standalone video generator, modern VAEs serve as robust perceptual compression stages that enable efficient training and modality fusion for multimodal video generation models, such as Latent Diffusion Models Hinton and Salakhutdinov (2006).

**VAE in Video Generation.** VAEs serve as a compression mechanism that transforms high-dimensional raw video data into compact latent representations. As shown in Figure 1b, a video VAE processes an input video sequence $x_{1:T}$ through a 3D encoder that incorporates both spatial and temporal convolutions to capture motion dynamics across frames. Video VAEs produce temporal parameters, temporal mean $\mu_{1:t}$ and temporal log-variance $\log \sigma_{1:t}^2$, that encode the full spatiotemporal structure of the video sequence.

The 3D encoder learns to compress the video into a probabilistic distribution in the latent space, where temporal dependencies are explicitly modeled. Following the reparameterization trick, a random noise sequence $\epsilon_{1:t}$ is sampled and combined with the temporal parameters to produce the spatiotemporal latent representation:

$$z_{1:T} = \mu_{1:t} + \sigma_{1:t} \odot \epsilon_{1:t} \tag{1}$$

where the subscript $1:t$ indicates that parameters and latents are computed for the temporal sequence rather than individual frames.

**VAE as Encoder in Multimodal Video Generation.** As illustrated in Figure 2, VAE is an important component in multimodal video generation architectures. In DiT-based approaches shown on the right, separate VAE encoders process video and audio modalities independently, producing modality-specific latent representations.

| Model / System | Architecture | Representative Notes | Release |
|---|---|---|---|
| *Proprietary Business Models* | | | |
| Google Veo 3.1 Wiedemer et al. (2025) | Unknown | Support 8 Second Video, I2VA, T2VA | May 2025 |
| OpenAI Sora 2 OpenAI (2025a) | Unknown | Max 10 Second Video, Portrait and Landscape cut, I2VA, T2VA | Sep 2025 |
| Grok 4 xAI (2025) | Unknown | I2V | July 2025 |
| Wan2.6 Wan et al. (2025) | Unknown | I2VA, T2VA | Dec 2025 |
| Kling 2.6 (Kuaishou) Kuaishou Technology (2025) | Unknown | I2VA, T2VA | Dec 2025 |
| *Open-Source Models* | | | |
| MM-Diffusion Ruan et al. (2023b) † | Decoupled U-Net | First Open-Sourced Multimodal Video Generation | Mar 2023 |
| OVI † Low et al. (2025) | DiT + synchronized audio-video | Native 4K generation at 50fps; open-source foundation model | Oct 2025 |
| LTX-2 (Lightricks)† Ha-Cohen et al. (2026) | DiT + synchronized audio-video | New Open-Surce SoTA | Jan 2026 |

Table 1: Representative Multimodal video generation models, their architectural paradigms, and release years. Open-source models are marked with †. I2VA represents Image to Video-Audio; T2VA represents Text to Video-Audio.

For the video stream, the Video VAE Encoder compresses raw video frames into video latents that capture visual content and motion patterns. Simultaneously, the Audio VAE Encoder transforms raw audio waveforms into audio latents that encode acoustic features and temporal dynamics. These parallel encoding pathways enable the model to handle heterogeneous data types within a unified framework.

## 2.2 Diffusion Architecture

Diffusion models Ho et al. (2020a) become a common robust generative architecture for multimodal video generation Low et al. (2025); HaCohen et al. (2026). The core idea is to train a neural network to reverse a gradual noising process, transforming samples from a simple noise distribution back into a complex data distribution such as images or videos Ho et al. (2022a). Diffusion models learn a sequence of denoising steps that progressively remove noise, ultimately yielding a high-quality sample.

Compared to VAEs, diffusion models differ in both training objective and generative process. VAEs rely on learning a low-dimensional latent space and optimize a variational lower bound, which often leads to blurry outputs due to the imposed likelihood assumptions Kingma and Welling (2013); Rombach et al. (2022). In contrast, diffusion models directly model the data distribution through iterative denoising, avoiding explicit latent bottlenecks and shows better visual aesthetic Dhariwal and Nichol (2021); Ho et al. (2022b). While diffusion models are typically slower at inference due to their multi-step sampling process Song et al. (2022),

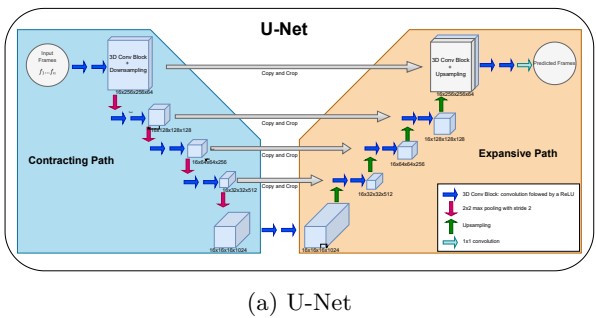

(a) U-Net

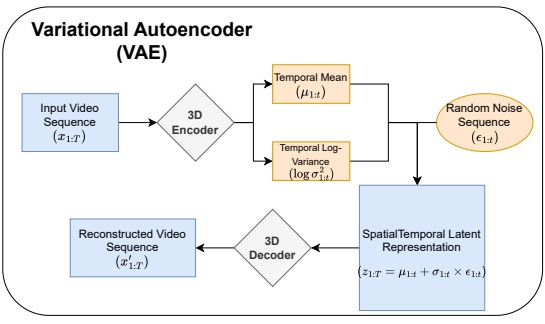

(b) Variational Autoencoder

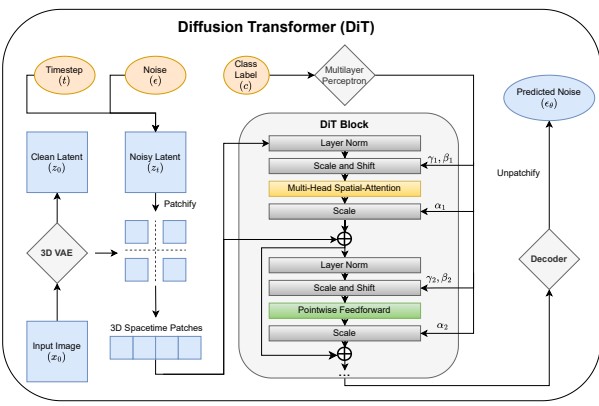

(c) Diffusion Transformer (DiT)

Figure 1: Core architectures in diffusion models: (a) U-Net with skip connections for iterative denoising, (b) Variational Autoencoder (VAE) for latent space encoding and decoding, and (c) Diffusion Transformer (DiT) replacing U-Net with transformer blocks for improved scalability.

they exhibit better visual output stability during training and better coverage of complex, high-dimensional data distributions Wang et al. (2025a); Yang et al. (2023). We discuss the block architectures and components of diffusion models and their representative models below.

### 2.2.1 U-Net

U-Net was one of the popular backbones for diffusion models, which uses a parametric function to predict noise or denoised signals at each diffusion timestep. Originally introduced for biomedical image segmentation Ronneberger et al. (2015), its encoder–decoder structure with skip connections has proven particularly effective for generative modeling. While modern architectures have increasingly shifted toward Diffusion Transformers (DiT) for multimodal video generation, U-Net laid the foundational groundwork for video generation Ho et al. (2022a) and joint audio-video synthesis Cheng et al. (2024) and remain relevant in certain contexts.

**Coupled U-Net for Joint Audio-Video Generation.** MM-Diffusion Ruan et al. (2023b) introduces the first joint audio-video generation framework using a coupled U-Net architecture. Rather than using a single network, MM-Diffusion uses two parallel U-Net subnets, one for video and one for audio, that jointly denoise Gaussian noise into aligned audio-video pairs. This coupled design establishes foundations in modern architectures: (1) modality-specific processing branches that respect the distinct characteristics of audio and video signals, and (2) cross-modal attention mechanisms that enforce alignment between generated modalities. Following U-Net-based works such as MM-LDM Sun et al. (2024) extends this paradigm by operating in a shared latent space to reduce computational costs while maintaining cross-modal consistency.

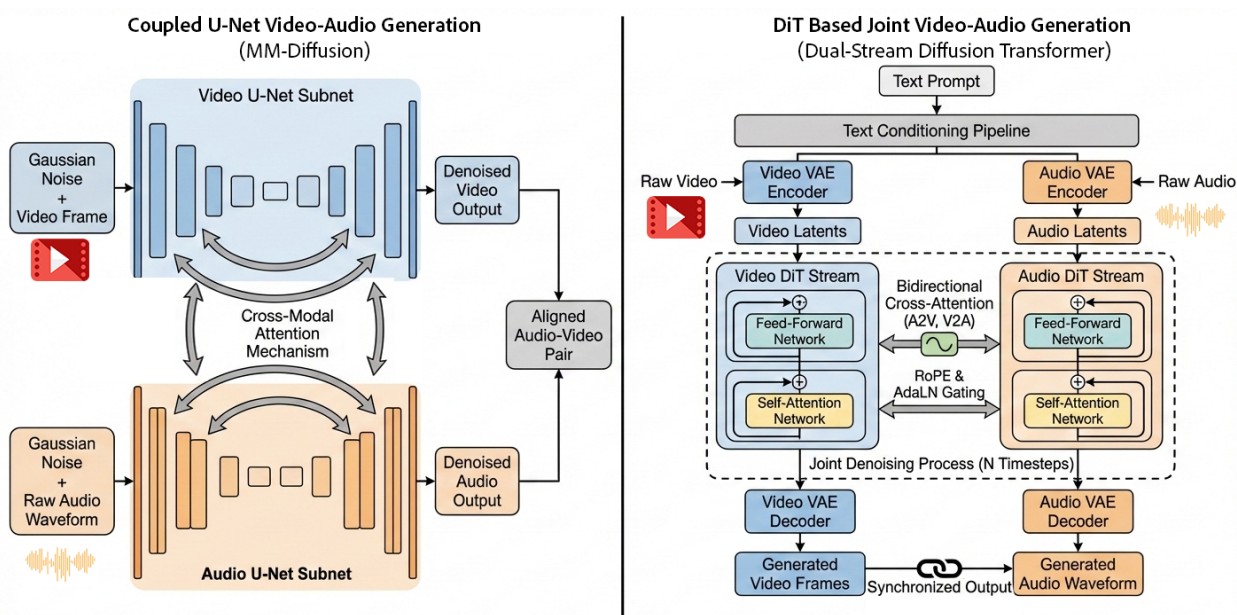

Figure 2: Evoluation of rchitectures for multimodal video generation in the open-source community.

**Audio Processing Representation in Multimodal U-Nets.** In joint audio-video frameworks, audio is typically represented as latent embeddings from pre-trained audio encoders, allowing the audio U-Net branch to process audio as 2D image-like representations Luo et al. (2023a); Liu et al. (2023a). MM-Diffusion processes raw audio waveforms through a dedicated U-Net subnet, while MM-LDM encodes audio into a shared latent space with video before applying the diffusion process.

**Limitations and Transition to DiTs.** Despite their foundational role, U-Net architectures face inherent limitations for joint audio-video generation Peebles and Xie (2023a). The locality of convolutional operations restricts long-range dependency modeling, which is crucial for maintaining semantic consistency across extended audio-video sequences Vaswani et al. (2017); Ma et al. (2024). Additionally, scaling U-Nets to higher resolutions and longer durations becomes computationally prohibitive Blattmann et al. (2023). These limitations motivate the transition toward DiT architectures, which show better scalability and more flexible *cross-modal attention mechanisms* Xu et al. (2025a).

### 2.2.2 Diffusion Transformer Backbones For Multimodal Video Generation

The Diffusion Transformer (DiT) is the successor to U-Net backbones, showing better scalability in model capacity, training data utilization, and generation fidelity Peebles and Xie (2023b). As illustrated in Figure 1c, DiT operates in a latent patch space: inputs are encoded by a pre-trained VAE, corrupted with noise, and processed as a sequence of tokens by a Transformer conditioned on timestep and semantic cues. The model learns to predict the noise via the standard denoising objective Ho et al. (2020b).

Unlike U-Net architectures, which rely on local convolutional inductive biases that struggle with distant dependencies Ronneberger et al. (2015); Ho et al. (2020b), Transformer backbones leverage self-attention to capture global spatiotemporal reasoning Peebles and Xie (2023b); Arnab et al. (2021). This global context is particularly vital for joint audio-video synthesis, where visual events (e.g., an explosion) must align perfectly with auditory signals (the sound) across long temporal windows. To manage the computational cost of this cross-modal alignment, architectures typically use either factorized spatial–temporal attention Ho et al. (2022c); Bertasius et al. (2021) for efficiency or full spatiotemporal attention Arnab et al. (2021); Tong et al. (2022) for maximal expressiveness.

**The Shift to Native Audio-Visual Synthesis.** By 2026, the industry has shifted toward native multimodal generation, driven by the realization that separate generation pipelines sever the natural correlation

between sight and sound. The release of OVI (2025) Low et al. (2025) and LTX-2 (2026) HaCohen et al. (2026) marked a turning point as the first open-source foundation models to generate synchronized video and audio using the DiT architecture, following proprietary successes like Sora 2 and Veo 3. Instead of training video and audios separately, existing open-source models use pre-trained video and audio encoders to pretrain jointly to fuse audio and video modalities to fit lip-sync (Figure 2). Below, we detail the specific architectural components these models use to unify audio and video generation.

**VAE encoders.** VAE encoders are generally used for training and conditional generation Diederik and Max (2019). In joint video-audio generation, there consists of a *video VAE encoder* Yu et al. (2024)and an *audio VAE encoder* Liu et al. (2024b). The Video VAE Encoder compresses raw video frames into compact video latents. A causal VAE architecture is used to to maintain temporal consistency. The Audio VAE Encoder Converts audio (via Mel Spectrogram) into audio latents. Also causal to preserve temporal structure.

**Text Conditioning Pipeline.** The text conditioning pipeline includes a Text Encoder, a pretrained language model or a multimodal encoder (like T5 Raffel et al. (2020) or CLIP Radford et al. (2021)) that tokenizes and encodes the text prompt; Feature Extractor refines the raw text encoder outputs into features better suited for conditioning; Text Connectors projects text embeddings to be compatible with both the audio and video streams

**Dual-Stream Diffusion Transformer Fusion** serves as the core mechanism enabling video and audio streams to communicate bidirectionally. Each stream contains self-attention for intra-modal coherence, text cross-attention (T2V, T2A) for semantic conditioning from the shared text embedding, and a feed-forward network. The key part is cross-modal communication: A2V cross-attention allows video queries to attend to audio keys/values, while V2A cross-attention does the reverse, enabling each modality to inform the other. This cross-attention uses Temporal 1D Rotary Positional Encoding (RoPE) Su et al. (2024) on queries and keys for temporal alignment across modalities, AdaLN (Scale, Shift) conditioned on each stream's diffusion timestep, and gating mechanisms to regulate cross-modal information flow.

**Joint Inference.** Both modalities start from independent Gaussian noise but denoise jointly in parallel over N shared timesteps. At each step, the dual-stream transformer processes both latents simultaneously, with bidirectional cross-attention allowing audio and video to communicate with each other. After denoising completes, separate VAE decoders reconstruct the final video frames and audio waveform.

### 2.2.3  Potential Future Architectural Design: Mixture of Experts and Autoregressive Generation

**Mixture of Experts (MoE).** As the video diffusion model component scales to billions of parameters, computational costs increase proportionally during both training and inference Wan et al. (2025); NVIDIA (2025). Mixture of Experts (MoE) architectures mitigate this challenge by introducing sparse activation, in which only a subset of model parameters is activated for a given input Shazeer et al. (2017).

While standard video-only MoE architectures route tokens based on spatial-temporal features (allocating experts to specialize in specific visual textures or motion dynamics Riquelme et al. (2021)), Audio-video MoEs must handle multimodal tokens with distinct sampling rates and semantic granularities, with the most recent works shown in Uni-MoE-2.0-omni Li et al. (2025b).

*Token-level MoE* Lepikhin et al. (2020); Fedus et al. (2022); Riquelme et al. (2021); Li et al. (2025c); Dai et al. (2024). In token-level MoE diffusion transformers, the standard feed-forward network (FFN) layers are replaced with multiple parallel expert networks and a learned routing function. For each input token $\mathbf{h}_i$, a router computes a distribution over experts and selects the top-$k$ experts $g_i$ (typically $k = 1$ or $k = 2$) to process that token:

$$g_i = \text{Top-}k\big(\text{Softmax}(W_r\mathbf{h}_i)\big),$$
$$\text{FFN}(\mathbf{h}_i) = \sum_{e \in g_i} p_{i,e} \cdot E_e(\mathbf{h}_i),$$

where $E_e$ denotes the $e$-th expert and $p_{i,e}$ are the routing weights. This design enables large model capacity while keeping per-token computation tractable. Token-level MoE is particularly effective when token complexity varies spatially or temporally, such as regions with intricate textures or complex motion patterns. Representative examples include SegMoE Ortigossa and Segal (2026) and Race-DiT Yuan et al.

(2025a), which demonstrate improved sample quality and reduced Floating Point Operations (FLOPs) through learned, dynamic routing.

***Timestep-level MoE*** Balaji et al. (2023); Zhuang et al. (2025); Cheng et al. (2025b). Beyond token-level routing, diffusion models show a natural form of multi-task structure across denoising timesteps: early timesteps (high noise phases) focus on global layout and motion planning, while later timesteps (low-noise phases) refine fine-grained appearance and temporal details. Timestep-level MoE uses this property by assigning different experts to distinct noise phases rather than routing individual tokens.

Formally, let $\epsilon_{\theta_e}(x_t, t)$ denote the denoising network of expert $e$ at timestep $t$. A hard-gated timestep MoE routes computation based solely on the diffusion timestep:

$$e(t) = \begin{cases} \text{high}, & t > t_{\text{switch}}, \\ \text{low}, & t \leq t_{\text{switch}}, \end{cases}$$

$$\epsilon_\theta(x_t, t) = \epsilon_{\theta_{e(t)}}(x_t, t).$$

Only a single expert is active at each timestep, resulting in sparse activation without introducing additional routing overhead. This formulation preserves inference cost per step while enabling strong specialization across denoising stages.

The state-of-the-art video-only generative model Wan 2.2 Wan et al. (2025) adopts such a timestep-specialized MoE design, maintaining separate model weights for high-noise and low-noise phases. During sampling, early denoising steps are handled by a high-noise expert that captures global structure and motion, while later steps are processed by a low-noise expert that refines textures and temporal consistency. This new design of MoE architecture improves video generation quality and efficiency compared to the non-MoE version (Wan2.1), also showing a promising architectureal design direction for adopting to multimodal video generation with MoE.

**Autoregressive Generation for Multimodal Video-Audio Synthesis.** Autoregressive (AR) generation has emerged as a promising paradigm for unifying different modalities (video, image, auido, text, etc) together due to its natural alignment with the sequential nature of temporal media Kondratyuk et al. (2024); ai et al. (2025), and it demonstrate strong scaling abilities in Large Language Models Yang et al. (2025a); OpenAI (2024), Vision-language models Li et al. (2025d); Bai et al. (2025), and unified models Deng et al. (2025); Xu et al. (2025b;c). For broader multimodal capabilities, Unified-IO2Lu et al. (2023), BAGEL Deng et al. (2025), EMMA He et al. (2025a), DeepSeek Nanus Pro Chen et al. (2025b) demonstrate that a single autoregressive transformer can be trained to understand and generate across text, images, audio, and action by tokenizing all modalities into a shared semantic space, though it processes these as separate generation tasks rather than jointly synthesizing video and audio. Similarly, Large World Model (LWM)Liu et al. (2024c) extends autoregressive transformers to million-token contexts for video understanding and generation but primarily focuses on the visual modality. For truly joint video-audio generation, diffusion-based approaches currently dominate Polyak et al. (2024); HaCohen et al. (2026); Low et al. (2025); Ruan et al. (2023a). However, the development of unified autoregressive architectures that can jointly understand and generate both video and audio in a single forward pass remains an open and promising research direction, with potential benefits including more natural temporal coherence and simplified training pipelines compared to multi-stage or separate-model approaches Kondratyuk et al. (2024).

## 3 Post-Training and Evaluation

Post-training and evaluation play critical roles in improving multimodal video-audio generation models on user-specified downstream tasks, where pre-trained base models fall short Liu et al. (2025b;c). We review several fundamental post-training methods for adapting video generation models to produce synchronized audio, including parameter efficient fine-tuning (PEFT) Mangrulkar et al. (2022), audio-visual alignment modules, attention manipulation, and ControlNet-based conditioning. We present the post-training introductions below.

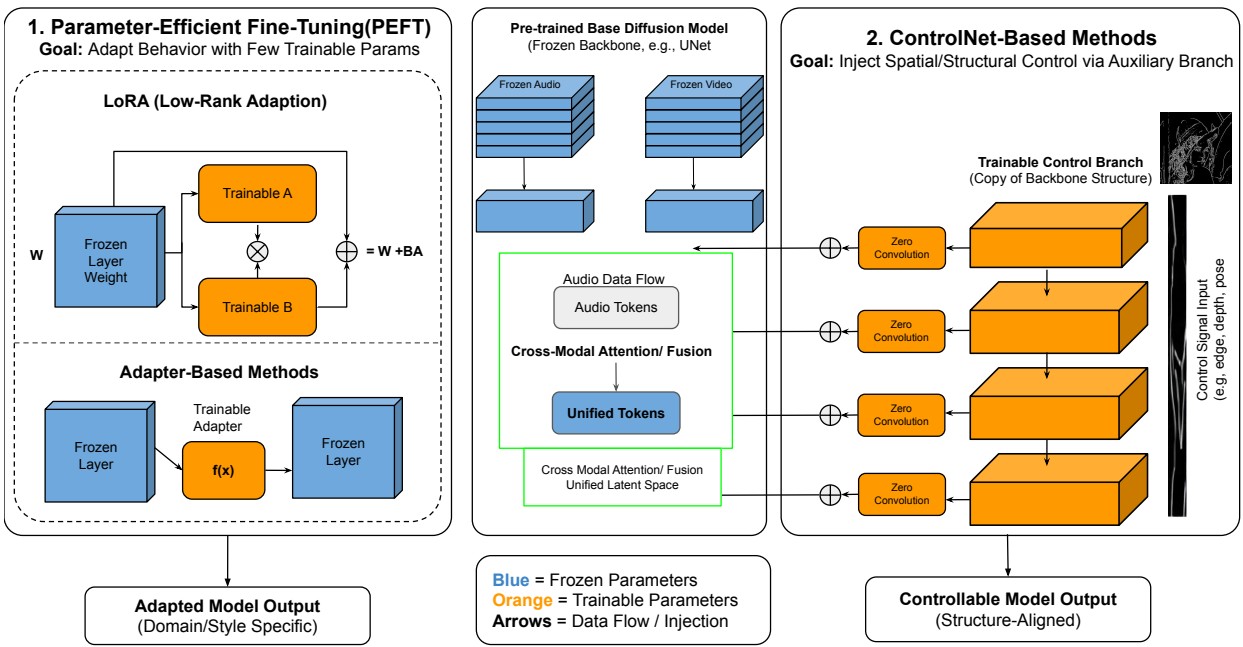

Figure 3: Exemplar post-training methods for multimodal video-audio generation.

## 3.1 Training Data Preparation

The post-training data preparation for multimodal video-audio generation requires carefully curated audio-video pairs with precise temporal alignment. For video-to-audio (V2A) tasks, training data consists of videos paired with their corresponding audio tracks, often with additional annotations such as sound event timestamps from datasets like AudioSet Strong Hershey et al. (2021), audio captions, or onset markers. For joint audio-video generation, synchronized multimodal data with aligned textual descriptions for both modalities is essential Cheng et al. (2025a); Zhao et al. (2025). The quantity of curated post-training data depends on the specific techniques: adapter-based methods like LoRA and semantic adapters typically require 100–1,000 video-audio pairs to generalize Zhang et al. (2024), while full-parameter fine-tuning may require millions of paired samples. Recent approaches leverage automated annotation pipelines using multimodal models to generate aligned video captions, audio captions, and speech transcriptions, ensuring temporal synchronization and semantic consistency Wang et al. (2025b).

## 3.2 Training-free Methods

Training-free methods operate during inference time to achieve audio-visual alignment without modifying base model weights Zhang et al. (2025b); Singer et al. (2025). These approaches enable novel multimodal capabilities on top of frozen pretrained models.

**Audio-Visual Guidance.** Training-free audio guidance methods steer video generation toward audio-synchronized outputs during the denoising process. By manipulating attention scores or injecting audio-derived conditioning signals at inference time, these approaches can enforce temporal alignment between generated video frames and audio events without additional training Xing et al. (2024a). Such methods are particularly useful for audio-to-video generation, where audio features guide the visual dynamics.

**Synchronization Guidance.** Recent work introduces synchronization guidance during sampling to strengthen audio-driven motion generation Song et al. (2025). By modifying the flow matching or diffusion loss to emphasize regions with large motion, and applying guidance that biases generation toward audio-

aligned temporal patterns, these methods improve lip synchronization and event timing without requiring model retraining.

### 3.3 Parameter-Efficient Fine-Tuning (PEFT)

Parameter-efficient fine-tuning (PEFT) Hu et al. (2022); Ruiz et al. (2022); Mangrulkar et al. (2022) adapts large pretrained models to multimodal audio-video tasks while introducing only a small number of trainable parameters. PEFT significantly reduces training cost and the risk of catastrophic forgetting, making it particularly attractive for adapting video or audio diffusion models to joint generation tasks.

**Low-Rank Adaptation (LoRA).** Low-Rank Adaptation Hu et al. (2022) injects trainable low-rank matrices into existing linear transformations, most commonly within attention layers. A weight matrix $W \in \mathbb{R}^{d \times d}$ is reparameterized as $W' = W + \Delta W$, where $\Delta W = BA$ with $A \in \mathbb{R}^{r \times d}$ and $B \in \mathbb{R}^{d \times r}$ ($r \ll d$). For multimodal generation, LoRA has been applied to adapt text-to-audio models for video conditioning, enabling efficient V2A fine-tuning. AV-DiT Wang et al. (2025c) demonstrates this approach by inserting LoRA layers into the projection modules of frozen attention blocks to bridge the domain gap between image-pretrained DiT and audio generation. Extensions apply LoRA to cross-modal attention layers, allowing video features to modulate audio generation while preserving the base model's audio quality.

**Adapter-Based Methods.** Adapter-based PEFT Xing et al. (2024b) introduces lightweight modules between layers of frozen backbones. For video-to-audio generation, semantic adapters process video features through trainable cross-attention layers that condition audio generation on visual content. FoleyCrafter employs a semantic adapter that utilizes parallel cross-attention layers to condition audio generation on video features, producing realistic sound effects semantically relevant to visual content. Temporal adapters further extend this paradigm by encoding timestamp conditions to achieve precise audio-video synchronization Zhang et al. (2024). AV-DiT Wang et al. (2025d) also incorporates bottleneck MLP adapters in parallel with feedforward networks to adapt image-based knowledge for audio modeling.

### 3.4 Audio-Visual Alignment Modules

A critical challenge in multimodal video-audio generation is achieving precise temporal and semantic alignment between modalities. Several specialized modules have been developed to address this challenge.

**Synchronization Modules.** Synchronization modules inject frame-aligned features to ensure precise audio-video temporal correspondence. MMAudio Cheng et al. (2025a) introduces a conditional synchronization module that aligns video conditions with audio latents at the frame level using aligned RoPE position embeddings and feature injection through adaptive layer normalization. The synchronization features are extracted using Synchformer Iashin et al. (2024), a self-supervised audio-visual desynchronization detector operating at 24 fps. This enables the model to capture fine-grained temporal correlations at millisecond granularity, which is critical since humans can perceive audio-visual misalignment as small as 25 milliseconds.

**Onset Detection and Timestamp Conditioning.** For video-to-audio generation, onset detectors predict when sound events should occur based on visual motion cues. FoleyCrafter Zhang et al. (2024) employs a timestamp detector that predicts sound and silence labels from video frames, trained with ground truth audio event timestamps from AudioSet Strong Hershey et al. (2021). The predicted timestamps are then encoded by a temporal adapter that follows the ControlNet design Zhang et al. (2023), injecting synchronization information into the audio generation backbone to ensure that generated sounds align with visual actions.

**Cross-Modal Feature Fusion.** Joint audio-video generation requires effective fusion of heterogeneous modality features. Approaches range from concatenating audio-video tokens in a shared latent space— UniForm Zhao et al. (2025) employs a unified single-tower DiT architecture to process concatenated audio-video tokens with task-specific noise schemes—to dual-branch architectures with cross-attention between parallel DiT streams. UniAVGen Zhang et al. (2025c) introduces asymmetric cross-modal interaction with bidirectional, temporally aligned cross-attention. Hierarchical encoders like Synchformer Iashin et al. (2024) capture fine-grained dynamic cues for temporal alignment, while semantic encoders (CLIP, SigLIP) provide global scene understanding for semantic consistency Cheng et al. (2025a); Shan et al. (2025).

**Contrastive Audio-Visual Pretraining (CAVP).** Diff-Foley Luo et al. (2023b) proposes contrastive audio-visual pretraining to learn temporally and semantically aligned features before training the diffusion model. CAVP uses a CLIP-like framework with both semantic contrast (maximizing audio-visual similarity within videos) and temporal contrast (emphasizing audio-visual synchronization within video segments). The CAVP-aligned visual features enable the latent diffusion model to capture subtler audio-visual correlations via cross-attention modules.

## 3.5 Attention Injection for Audio-Visual Control

Attention injection modifies attention mechanisms to introduce audio or video conditioning signals while preserving backbone weights. Unlike PEFT that adapts through weight updates, attention injection operates directly in the attention space Yuan et al. (2025b); Cai et al. (2025a).

**Cross-Attention for Modality Conditioning.** A common approach introduces cross-attention layers where audio latents attend to video features (or vice versa). FoleyCrafter Zhang et al. (2024) integrates parallel cross-attention layers alongside text-based attention, allowing audio generation to be conditioned on video features without compromising text-to-audio capabilities. For audio-to-video generation, Syncphony Song et al. (2025) injects audio features via cross-attention with RoPE to enable audio-motion alignment on top of a DiT architecture.

**Joint Self-Attention.** Joint audio-video generation models often employ unified self-attention over concatenated audio and video tokens. MMAudio Cheng et al. (2025c) uses multimodal transformer blocks where video, text, and audio latents jointly interact, allowing bidirectional information flow between modalities during the denoising process. UniForm Zhao et al. (2025) similarly processes audio and video tokens within a unified latent space using a shared DiT, enabling the model to learn implicit correlations between visual dynamics and audio characteristics.

**Attention Feature Injection.** Some methods inject external attention features—such as precomputed audio embeddings or onset-aligned temporal features—into existing attention layers. Rather than adding new layers, these approaches modify attention computation by blending or biasing query-key affinities. MMAudio Cheng et al. (2025a) uses this approach for temporal synchronization, where Synchformer-derived features are injected through adaptive layer normalization to enforce audio-video alignment at specific timestamps. AV-DiT Wang et al. (2025d) facilitates feature interaction between audio and visual modalities by pooling video tokens temporally and concatenating them with audio tokens in a shared attention block augmented with LoRA.

## 3.6 ControlNet-Based Methods for Audio-Visual Generation

ControlNet-based approaches Zhang et al. (2023); Chen et al. (2025c); Gu et al. (2025) extend pretrained diffusion models with conditional branches to enable fine-grained control over multimodal generation. For video-to-audio synthesis, ControlNet augments audio generation backbones with video-conditioned control networks.

**Temporal ControlNet for V2A.** FoleyCrafter's Zhang et al. (2024) temporal adapter follows the ControlNet design, duplicating the UNet encoder structure and introducing zero-initialized connections that inject temporal control features into the audio generation process. The adapter takes timestamp masks as input and adds residual control signals to the original UNet, enforcing precise temporal alignment between generated audio and video events. During training, only the replicated UNet blocks are updated using the same optimization objective as the diffusion model.

**Multi-Stream Temporal Control.** For complex audio-visual scenarios involving speech, sound effects, and music, multi-stream ControlNet architectures process different audio components separately. MTV Weng et al. (2025) introduces a Multi-Stream Temporal ControlNet with interval streams for speech and effects tracks (controlling lip motion and event timing) and holistic streams for music (controlling visual mood). The interval stream employs interval interaction blocks to understand each track individually, while the holistic stream extracts features using a holistic context encoder that serves as style embeddings applied uniformly to all frames.

**Video Feature Injection via ControlNet.** ControlNet can embed video characteristics into text-to-audio synthesis by processing visual features through a control branch. FoleyCrafter Zhang et al. (2024) extracts video frames using pretrained visual encoders and injects the resulting features into the audio diffusion backbone. HunyuanVideo-Foley Shan et al. (2025) extends this by using Representation Alignment (REPA) to align intermediate DiT representations with frame-level audio features from pretrained self-supervised models, enhancing both semantic and acoustic modeling.

## 3.7 Common Evaluation Practices

Evaluating joint-video-audio generation is hard, with two main common evaluation practices for joint video-audio generation models: *quantitative* and *qualitative*. Quantitative evaluations for text-to-video-audio (T2VA), video-to-audio (V2A), and related multimodal generation tasks use automated metrics to assess the quality of generated content across both modalities Ruan et al. (2023a); Kilgour et al. (2019a); Unterthiner et al. (2019); Wu* et al. (2023), while qualitative evaluations rely on human judgments and rubrics to rate perceptual quality, synchronization, and semantic coherence Luo et al. (2023b); Huang et al. (2024a; 2025b); Zheng et al. (2025); Liu et al. (2023b).

Figure 4: Multimodal Evaluation Common Practices.

### 3.7.1 Quantitative Evaluations

Joint video-audio evaluation requires metrics that assess each modality independently as well as their cross-modal alignment. These metrics are in three categories: video quality, audio quality, and audio-visual alignment.

**Video Quality Metrics.** Common metrics include *Fréchet Video Distance (FVD)*, a reference-based matric which measures the distribution distance between generated and real video features using I3D network embeddings pretrained on Kinetics Kay et al. (2017), capturing both spatial quality and temporal coherence Unterthiner et al. (2019). *CLIPScore* Hessel et al. (2021) is a reference-free metric that computes the cosine similarity between CLIP Radford et al. (2021) visual embeddings of generated video and the textual embedding of the input prompt, thereby measuring text-video semantic alignment. Originally proposed for image captioning evaluation, it has been widely adopted in video generation by averaging frame-level CLIP similarities across all frames of a generated video.

*VBench series* Huang et al. (2024a; 2025b); Zheng et al. (2025) provide a more fine-grained evaluation by decomposing video generation quality into many dimensions. Representative and popular dimensions

include e.g, *Dynamic Degree*, which measures the extent of motion present in the generated video; *Motion Smoothness*, measures the motion priors of a video frame interpolation model to assess whether generated motions are physically plausible; *Aesthetic Quality* measures the artistic and beauty value perceived by humans towards each video frame using the LAION aesthetic predictor Wu* et al. (2023).

VBench++Huang et al. (2025b) extends this framework to image-to-video evaluation and model trustworthiness assessment, while VBench-2.0Zheng et al. (2025) introduces 18 additional dimensions targeting intrinsic faithfulness, including physics-based realism, commonsense reasoning, and human fidelity, leveraging VLM-based evaluation pipelines to assess capabilities where earlier metrics have begun to saturate Li et al. (2025e); Guan et al. (2024).

**Audio Quality Metrics.** Fréchet Audio Distance (FAD) compares generated and reference audio distributions in an embedding space, typically using VGGish features Kilgour et al. (2019a). Kullback-Leibler (KL) divergence measures the distribution similarity between generated and target audio classification outputs. Inception Score (IS) adapted for audio evaluates sample diversity and quality. Kernel Audio Distance (KAD) addresses FAD's Gaussian assumption limitations through an MMD-based approach Chung et al. (2025a). For text-conditioned generation, CLAP Score measures text-audio alignment via cosine similarity in the CLAP embedding space Wu* et al. (2023). Recent advancements (2024–2025) have introduced reference-free and model-based metrics to address the reliance on ground-truth datasets. PAM leverages Audio-Language Models to score quality via text prompting Deshmukh et al. (2024), while Audiobox Aesthetics decomposes quality into specific axes like *Production Quality* and *Content Enjoyment* Vyas et al. (2023). Additionally, MAUVE Audio Divergence (MAD) has emerged as a non-Gaussian alternative to FAD for better distribution profiling Zhang et al. (2025d).

**Audio-Visual Alignment Metrics.** Evaluating the synchronization and semantic coherence between generated audio and video requires specialized metrics that assess whether sounds correspond to visible events, occur at appropriate times, and originate from correct spatial locations Girdhar et al. (2023a); Goncalves et al. (2024b); Cheng et al. (2025a). AV-Align Yariv et al. (2023a) measures semantic correspondence between audio and video streams, while DeSync Feng et al. (2025a) quantifies temporal misalignment in seconds using the Synchformer model. For cross-modal evaluation in a shared embedding space, ImageBind Score (IB) Girdhar et al. (2023b) computes the cosine similarity between audio and video representations projected into ImageBind's joint embedding space. FAVD Kilgour et al. (2019b) extends the Fréchet distance framework to joint audio-visual features, capturing distributional similarity of generated audio-video pairs against real data. Finally, Spatial AV-Align Yariv et al. (2023a) evaluates spatial coherence by combining object detection with sound event localization and detection (SELD) to verify that generated sounds originate from the correct locations in the visual scene.

### 3.7.2 Qualitative Evaluation

Human evaluation is essential for joint video-audio generation, as automatic metrics often fail to capture perceptual synchronization quality and semantic coherence across modalities Huang et al. (2024a); Liu et al. (2025c); Ruan et al. (2023a). Evaluations typically involve in two paradigms:

**Overall Preference.** Annotators select the better sample between model outputs or rate overall quality on a Likert scale (typically 1–5), following the Mean Opinion Score (MOS) protocol standardized in ITU-T P.800 MET. For audio-visual content, annotators assess the combined experience of seeing and hearing the generated output.

**Multi-Aspect Scoring.** Quality is decomposed into modality-specific and cross-modal dimensions (Table 2).

The PEAVS framework Goncalves et al. (2024b) provides a comprehensive protocol for perceptual evaluation of audio-visual synchrony, covering temporal offsets, speed variations, and content-level alignment. For stereo audio generation, evaluators additionally assess whether spatial audio positioning corresponds to object locations in the visual scene Zhou et al. (2020).

| Paradigm | Category | Metrics / Aspects |
|---|---|---|
| Quantitative | Video Quality | FVD Unterthiner et al. (2018), CLIPScore Hessel et al. (2021), VBench series Huang et al. (2024b) |
| | Audio Quality | FAD Kilgour et al. (2019c), KL Divergence Kullback and Leibler (1951), KAD Chung et al. (2025b), CLAP Score Elizalde et al. (2022), PAM Deshmukh et al. (2024), Audiobox Aesthetics Tjandra et al. (2025) |
| | AV Alignment | AV-Align Yariv et al. (2023b), DeSync Zhou et al. (2025), ImageBind Score Girdhar et al. (2023b), FAVD Mo et al. (2024), Spatial AV-Align Shimada et al. (2024a) |
| Qualitative | Protocol | Overall Preference (MOS) itu (2016), Multi-Aspect Scoring |
| | Video Aspects | Visual Fidelity, Temporal Coherence, Motion Realism Huang et al. (2024b); Han et al. (2025) |
| | Audio Aspects | Audio Quality, Sound Relevance, Audio Diversity Kreuk et al. (2023); Vinay et al. (2022) |
| | Cross-Modal | AV Synchronization Goncalves et al. (2024c), Semantic Coherence, Spatial Consistency Shimada et al. (2024b) |
| | Prompt | Text Alignment Han et al. (2025) |

Table 2: Evaluation Framework for Joint Video-Audio Generation

## 4 Applications and New Research Directions

With the emerging capabilities and scalability of joint video-audio generation models Wiedemer et al. (2025); HaCohen et al. (2026), multimodal content creation is entering a new phase where visual and auditory elements are produced synchronously rather than as separate post-production steps. Starting in 2026, the creative and commerce landscape is moving beyond the *uncanny valley* of silent or post-dubbed generated video into an era where synchronized soundscapes, dialogue, ambient audio, sound effects, and music, are generated alongside the visual stream as a unified output OpenAI (2025a); HaCohen et al. (2026); xAI (2025). The current market also reflects this shift. Proprietary models such as Sora 2 OpenAI (2025b), Veo 3.1 Google AI Studio (2025); Google DeepMind (2025), Wan2.6 AtlasCloud (2025); Higgsfield (2025), and Grok 4 xAI (2026) have begun integrating native audio generation pipelines, while creator platforms like Kling AI Kling AI (2025), Runway Gen-4 Runway (2025a; 2026), Pika Pika (2024; 2026), and Doubao Yahoo Finance (2024); WIRED (2025) on TikTok increasingly provide joint audiovisual outputs that eliminate the traditional separation between video editing and sound design. Creator-friendly interfaces such as ComfyUI ComfyUI Blog (2025; 2026); gitmylo (2026) are similarly incorporating audio-aware workflows. In this section, we summarize popular applications of joint video-audio generation and how the addition of synchronized audio transforms use cases that were previously limited to silent video for both personal users and the industry. We then survey active research and engineering directions specific to the audiovisual setting, including temporal audio-visual synchronization, spatially grounded sound generation, and multimodal coherence at scale.

### 4.0.1 Personal User Applications

**Social Media Content Creation and Entertainment.** One of the most widespread applications is short-form video generation for social media platforms. Users can now generate engaging audiovisual content directly from text prompts or reference images, without the need for cameras, actors, sound recording equipment, or complex editing software Zheng et al. (2024). OpenAI's *Sora 2* OpenAI (2025a), Tiktok's Doubao Yahoo Finance (2024) Kuaishou (Kling AI) Kuaishou (2024) allow users to create and remix stylized short clips with matched soundtracks from images or text prompts and post to their own media platform. The addition of audio-driven animation to social media platforms, where an audio clip can drive facial expressions and lip movements in the animated photograph, deepens the sense of presence and realism, while simultaneously raising ethical considerations around consent and authenticity Rosenberg (2025).

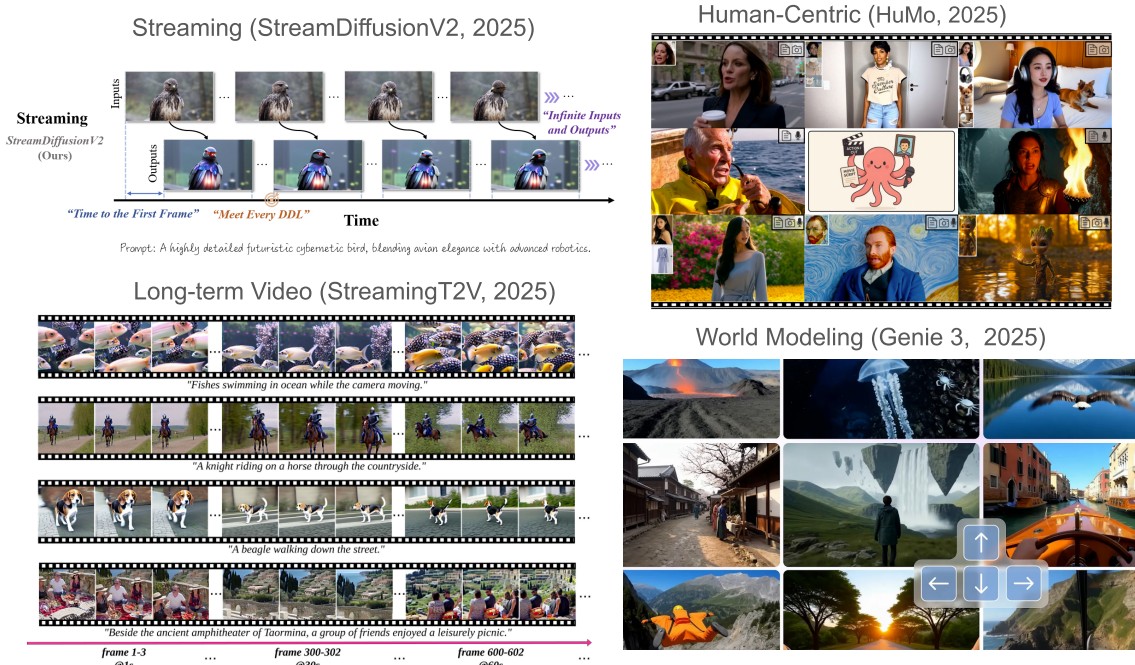

Figure 5: Main Stream Multimodal Video Generation Research Areas. Streaming video generation to generate real-time videos; Human-centric multimodal video generation that generates mainly based on human characters with voices; long-term video generation where to maintain temporal consistency for long generation; world modeling for generating simulation of real-world and sound.

**Multimodal Video Personalization and Avatar Creation.** Multimodal video generation has enabled sophisticated personalization through virtual avatars and digital humans. Recent advances in audio-conditioned generation allow these avatars to speak and emote with natural lip synchronization and co-speech gestures driven directly from audio input, reducing the need for manual motion capture or animation rigging. Audio-driven portrait animation methods such as *Hallo* Xu et al. (2024a), *EchoMimic* Chen et al. (2024); Meng et al. (2024), and *EMO* Tian et al. (2024; 2025) allow users to create talking avatars from a single photograph, generating synchronized lip movements and facial expressions from an audio track. ByteDance's *OmniHuman-1* Lin et al. (2025a) substantially advances this paradigm by supporting full-body audio-driven animation, including talking, singing, and gestural co-speech motion, from a single reference image and an audio signal.

### 4.0.2 Commercial Applications of Video Generation

Products and models are being developed, but they cannot sustain growth until they are deployed in real-world use cases, generate revenue, and receive feedback from actual users Maslej et al. (2024). As joint video-audio generation models advance, commercial adoption is accelerating across advertising, media production, and enterprise workflows, with native audio capabilities emerging as a key differentiator that reduces post-production overhead and enables fully immersive audiovisual content from a single generative pipeline.

**Advertisement and Marketing.** Google demonstrated this capability by creating full TV commercials with *Veo 3*, including a holiday advertisement that aired on television and in cinemas, with synchronized dialogue, sound, and music generated natively alongside the visuals Sullivan (2025).

**Film and Video Production.** A representative example is at Adobe MAX 2025, Adobe introduced *Firefly* as an all-in-one creative AI studio with multimodal generation spanning image, video, and audio Adobe Newsroom (2025). The platform includes *Generate Soundtrack* (powered by the commercially safe Firefly

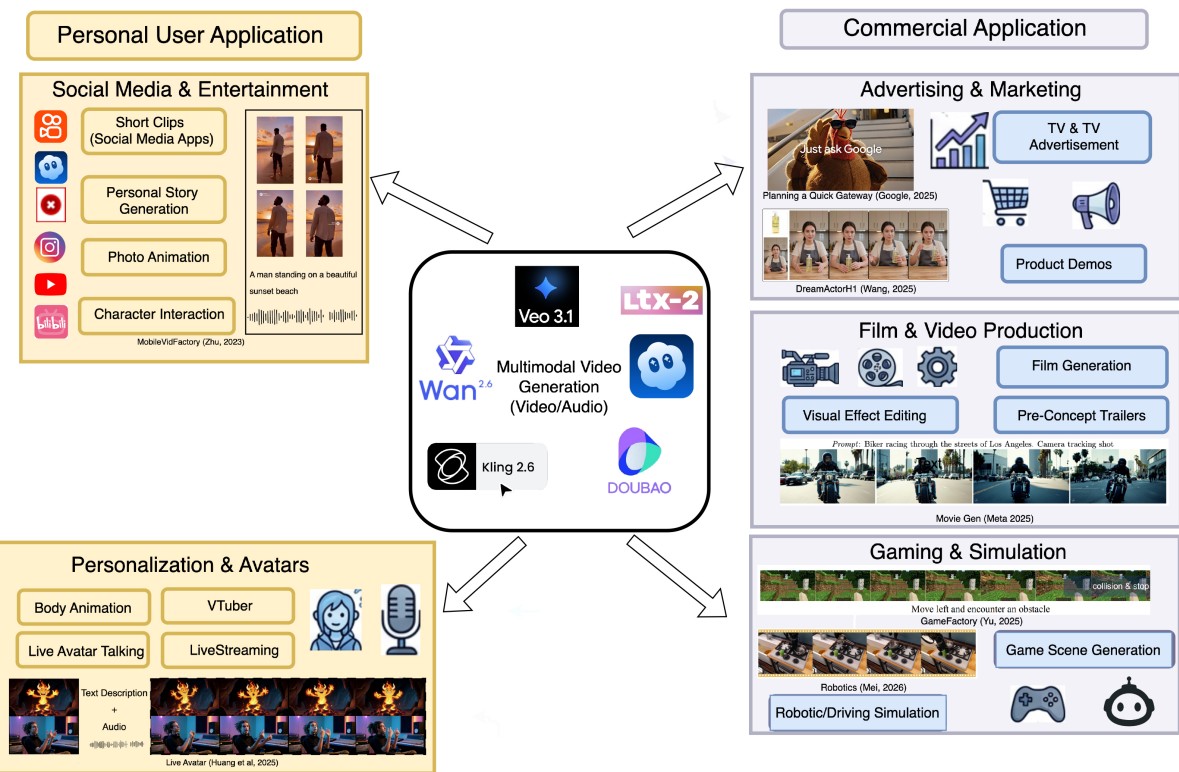

Figure 6: Widely used applications of multimodal video generations. They mainly focus on personal and business applications.

Audio Model) for creating fully licensed, studio-quality instrumental tracks synchronized to video footage, and *Generate Speech* for crystal-clear voiceovers Chedraoui (2025).

**Gaming and Interactive Entertainment.** *ElevenLabs* ElevenLabs (2025) provides AI voice generation for NPCs and player characters, supporting real-time text-to-speech with low latency, voice cloning for consistent character voices, which enables games to feature adaptive, personalized dialogue without pre-recorded voice acting for every scenario.

The addition of synchronized audio to video generation pipelines marks a fundamental shift from silent visual generation to complete audiovisual content creation, reducing production overhead and enabling new creative possibilities across commercial applications.

## 5 Active Research Areas in Multimodal Video Generation

With the advancement of multimodal video generation, there exist many active research areas to explore. Specifically, these areas include Streaming Multimodal Video Generation, Human-Centric Multimodal Generation, and Long Multimodal Video Generation. We summarize the most recent active research directions and their implications for joint video-audio synthesis below.

### 5.1 Streaming Multimodal Video Generation

Streaming video generation produces frames incrementally in real time with strict latency and memory constraints Kodaira et al. (2025a). Extending this paradigm to multimodal generation requires maintaining audio-visual synchronization during incremental synthesis, which is essential for live avatars, interactive

agents, and real-time content creation Huang et al. (2025c). However, this introduces new challenges such as *causal* context degrades generation quality overtime; *memory efficiency* becomes a bottleneck for maintaining separate Key-Value (KV) caches for long-duration video and audio streams; *cross-modal alignment* without future look-ahead is difficult, where minor latency variations between audio and video streams can cause noticeable synchronization drift.

**Causal temporal modeling** refers to autoregressive generation where each frame at time $t$ is predicted from past frames and latent states without access to future context, enabling on-the-fly, low-latency synthesis Yan et al. (2021). **CausVid** adapts a pretrained bidirectional video diffusion model into a causal autoregressive transformer, introducing distribution matching distillation to compress a slow bidirectional teacher into a fast causal student Yin et al. (2025). **MotionStream** extends causal autoregressive video generation to interactive, motion-controlled streaming using sliding-window causal attention with attention sinks and KV cache rolling Shin et al. (2025). **StreamDiffusion** provides a real-time diffusion pipeline optimized for interactive generation Kodaira et al. (2025b); Feng et al. (2025b). For multimodal streaming, these causal architectures can be extended to jointly generate synchronized audio tokens alongside video frames, requiring cross-modal attention mechanisms that operate under strict causality constraints.

**Temporal consistency without teacher forcing.** Streaming models must self-condition on their own previous outputs, leading to drift errors. **Self-Forcing** addresses this by simulating the inference process during training, conditioning each frame on its own previously generated outputs Huang et al. (2025d). **Reward Forcing** introduces reward-guided distribution matching distillation with an EMA-Sink mechanism for motion consistency Lu et al. (2025). **LongLive** aligns training with long-sequence inference to reduce temporal drift Yang et al. (2025b). **VideoGPA**distills scene-level geometric priors from a feedforward geometric foundation model to mitigate temporal drift Du et al. (2026). For multimodal generation, these self-conditioning strategies must account for both visual drift and audio-visual desynchronization over extended generation.

**Interactivity and Reactivity.** Interactive streaming responds to real-time user inputs. **PersonaLive** enables real-time portrait animation for live streaming Li et al. (2025f). **LiveAvatar** supports interactive, infinite-length avatar video generation conditioned on live audio or dialogue inputs Huang et al. (2025c). This existing research sshowcases the ability of integration of audio conditioning in streaming generation in the current pure video generation pipeline, pointing toward fully joint audio-video streaming synthesis.

## 5.2 Human-Centric Multimodal Video Generation

Human-centric video generation produces realistic imagery of human subjects conditioned on signals such as audio, pose, or text Hu et al. (2024a); Xu et al. (2023). This domain is inherently multimodal, as human video often requires synchronized speech, environmental sounds, and background music Xu et al. (2024b); Siyao et al. (2022). However, multimodal human-centric generation faces challenges such as *Fine-grained synchronization*, *temporal consistency*, and *computational efficiency*.

**Face Animation.** Face animation synthesizes realistic facial motion, such as lip articulation, expressions, and head movement, conditioned on driving signals including audio, text, or motion cues . Audio-driven methods such as MultiTalk Kong et al. (2025), InfiniteTalk Yang et al. (2025c), and FantasyTalking Wang et al. (2025e) map speech signals to mouth shapes and facial motion. More recent approaches including StableAvatar Tu et al. (2025), LongCat-Avatar Meituan LongCat Team (2025), and KlingAvatar 2.0 Kling Team et al. (2025) extend to infinite-length generation, addressing temporal drift and identity degradation. Streaming systems such as PersonaLive Li et al. (2025f), AnyTalker Zhong et al. (2025), and LiveAvatar Huang et al. (2025e) push toward real-time, low-latency generation with live audio input. Expressive models like OmniHuman-1 Lin et al. (2025b) and Sonic Ji et al. (2025) incorporate global audio perception and multimodal conditioning for richer facial dynamics Ji et al. (2025); Lin et al. (2025b).

**Pose Animation.** Pose animation generates full-body human motion videos from pose sequences or control signals. Methods such as Wan-Animate Cheng et al. (2025d), SCAIL Yan et al. (2025a), and Follow Your Pose map target poses to realistic human videos, while SteadyDancer Zhang et al. (2025e) and AnimateDiff Guo et al. (2024) enhance temporal fidelity through diffusion-based architectures. Integrating audio with pose animation enables music-driven dance generation and speech-driven gesture synthesis, requiring models to

learn correlations between audio rhythm/prosody and body movement dynamics Yi et al. (2023); Tseng et al. (2023).

**Customization.** Customization enables controllable identity, appearance, and motion in generated videos. Models such as HuMo Chen et al. (2025d), FFGo Wu et al. (2025), and Stand-In Xue et al. (2025) transfer target identities onto video sequences, while HunyuanCustom Hu et al. (2024b), HyperMotion Xu et al. (2025d), and Phantom Liu et al. (2025d) support fine-grained control over gestures and motion. For multimodal generation, customization extends to voice cloning and audio style/identity control, enabling consistent audio-visual identity across generated content Qiang et al. (2026); He et al. (2025b).

## 5.3 Long Multimodal Video Generation

Long video generation produces temporally coherent videos extending over minutes to unbounded lengths Elmoghany et al. (2025). For multimodal content, this requires maintaining audio-visual synchronization and semantic consistency over extended durations. However, scaling generation to extended durations introduces critical bottlenecks with *Temporal drift* and *computational efficiency* as primary concerns.

**Single-Shot Generation.** Single-shot methods generate long videos frame-by-frame in a continuous manner using latent caching, rolling attention windows, or self-conditioning. Representative works include FramePack Zhang et al. (2025f), StreamingT2V Henschel et al. (2025), Rolling Forcing Liu et al. (2025e), LongLive Yang et al. (2025b), SVI Li et al. (2025g), Self-Forcing++ Cui et al. (2025), LongVie Gao et al. (2025), LongCat-Video Meituan LongCat Team et al. (2025), FreeLong Lu et al. (2024), and Infinity-RoPE Yesiltepe et al. (2025). Extending these to multimodal generation requires joint latent caching for both video and audio streams, with synchronization mechanisms that prevent cross-modal drift over long sequences.

**Multi-Shot Generation.** Multi-shot methods divide videos into segments, synthesizing each with context-aware conditioning for cross-boundary coherence An et al. (2025). Examples include HoloCine Meng et al. (2025), Mixture of Contexts Cai et al. (2025b), and StoryMem Zhang et al. (2025g), which use memory mechanisms for consistent narrative and style. For multimodal content, multi-shot approaches enable scene-aware audio generation where audio characteristics (ambient sounds, music, dialogue) can shift naturally across scene boundaries while maintaining global coherence Zhang et al. (2025h).

**Agentic Storytelling.** Recent agentic systems improve long-horizon video alignment via iterative planning and critique: VISTA Long et al. (2025) refines prompts in a loop using temporal plans, tournament selection, and specialized critics; VideoAgent Soni et al. (2024) reduces hallucinations in video plans for robot control via self-conditioning consistency and environment feedback; AutoMV Tang et al. (2025) coordinates music analysis with script/director/verifier agents to generate coherent full-length MVs and benchmark them against human experts; and ScripterAgent Mu et al. (2026) translates dialogue into executable scripts (ScriptBench) that guide cross-scene generation, evaluated with CriticAgent and a visual–script alignment metric.

## 5.4 Interactive Multimodal Video Generantion and World Models

World models Ha and Schmidhuber (2018) encompass a broad class of approaches that aim to capture and simulate the underlying dynamics of the real world at varying levels of abstraction. Typically, a world model maintains an internal representation of the environment state, receives actions from decision-making agents, and predicts the subsequent state conditioned on those actions. While early world models focused primarily on visual observations, real-world perception is inherently multimodal—audio carries rich information about environmental properties that are often complementary to visual cues, including spatial location of sound sources, acoustic characteristics of physical spaces, and temporal evolution of auditory events.

Based on differences in goals and methodologies, world models can be broadly divided into two categories Huang (2025): **(a) Representation World Models**, which learn abstract, semantic-level representations to predict physical events, often embedded within LLM/VLM or agentic frameworks Zhang et al. (2025i); Bolton et al. (2025); and **(b) Generative World Models**, which represent the world state as a detailed description of the physical environment, functioning as high-fidelity simulators that explicitly generate

future world states. In this survey, we primarily focus on the latter category, with emphasis on multimodal generation capabilities.

**Spatial and texture awareness in audio–visual generation.** A key challenge in interactive multimodal generation is producing audio that matches physical cues—spatial layout, material/texture, and resonance. Recent work makes steady progress: ELSA Devnani et al. (2024) learns spatially grounded text–audio embeddings for open-vocabulary retrieval and language-based 3D localization; Visual Acoustic Fields Li et al. (2025h) couples 3DGS with diffusion to generate and localize impact sounds in 3D; xRIR Liu et al. (2025f) generalizes RIR prediction across rooms using depth geometry plus a few reference RIRs; AV-DAR Jin and Gao (2025) renders acoustics via differentiable beam tracing guided by multi-view visuals; and ViSAGe Kim et al. (2025) synthesizes first-order ambisonics from silent video with directional guidance, outperforming two-stage spatialization pipelines.

**Audio-Visual World Models.** Recent work extends world models from vision-only to synchronized audio generation. **Audio-Visual World Models (AVWMs)** target action-conditioned simulation of joint audio-visual dynamics with reward prediction; AV-CDiT Wang et al. (2025f) uses a diffusion transformer with modality experts and stagewise training for balanced multimodal forecasting, enabling coherent futures for embodied AI (e.g., audio-visual navigation). Zhang and Gienger (2025) proposes a latent flow-matching world model that predicts future audio for temporally grounded planning, improving manipulation under in-the-wild sounds and music where rhythmic dynamics matter. GWM-1 Runway (2025b) is a real-time, action-conditioned "general world model" (Gen-4.5) that generates controllable frame-by-frame simulations across explorable worlds, avatars, and robotics. MovieGen Polyak et al. (2024) combines a 30B video model with a 13B audio model to produce synchronized ambient/Foley/music, learning physical and perceptual audio-visual links for realistic generation. Veo3 Veo-Team (2024) adds audio synthesis to video generation, reflecting the broader trend toward unified audio-visual simulation.

**Integration with Embodied Intelligence.** For embodied agents operating in real environments, multimodal world models provide critical capabilities beyond visual prediction. Audio signals enable agents to perceive occluded objects, estimate room acoustics, and anticipate events before they become visible Somayazulu et al. (2024). Joint MLLM-WM architectures are emerging where MLLMs provide semantic reasoning and task decomposition while WMs offer physics-aware simulation including audio-visual dynamics Feng et al. (2025c). Such multimodal world models support planning in environments where audio cues are essential—for example, navigating toward sound sources, predicting acoustic consequences of actions, or generating appropriate audio feedback in interactive simulations Wang et al. (2025f).

**Challenges and Future Directions.** Extending world models to multimodal audio-visual generation presents several challenges: (1) ensuring precise temporal synchronization between visual dynamics and audio events at millisecond granularity; (2) modeling the physical acoustics of environments including reverberation, occlusion, and spatial audio; (3) generating semantically appropriate sounds for novel visual concepts not seen during training; and (4) scaling multimodal world models to support long-horizon generation while maintaining audio-visual coherence. Addressing these challenges will be essential for building world simulators that capture the full multisensory nature of physical reality.

## 6 Limitation and Challenges

The field of video-audio generation is experiencing a rise in popularity and rapid advancement. State-of-the-art models, architectures, and applications are advancing at an unprecedented pace. While this review highlights pioneering works and outlines potential research directions for multimodal integration, it cannot claim to be exhaustive given the field's hype. Instead, we aim to provide a foundational overview, concluding with a critical discussion of the current limitations and remaining challenges in multimodal video generation.

**Evaluation.** Critical challenges still remain in multimodal video generation evaluations. There is no universally adopted metric for audio-visual synchrony or semantic alignment, leading models to report diverse and sometimes incomparable metrics. Many audio metrics (e.g., FAD Kilgour et al. (2019a)) operate on downsampled mono audio, making them insensitive to high-frequency content and stereo characteristics Liu et al. (2023a). Furthermore, the correlation between automatic metrics and human perception varies significantly

across content types and generation quality levels, necessitating continued reliance on human evaluation for comprehensive assessment Hua et al. (2025).

**Model Efficient Deployment and Latency.** As multimodal video generations are going to be deployed and come into our daily life, massive users will use the service for various purposes Chakraborty and Biswal (2025). Especially for interactive and real-time interaction. In such cases, challenges remain for multimodal generation, where high inference latency disrupts the experience of human-AI interaction, and the computational burden of processing synchronized audio-visual streams necessitates aggressive model compression (e.g., quantization, distillation) that often degrades temporal coherence Chern et al. (2025); Chen et al. (2025e).

**Modality Fusion and Unified Generation.** Although a line of works already studied the fusion and interaction between different modalities (text, video, audio) and try to unify those modalities together for synchronous generation and understanding, many challenges remain for studying the relationship between different modalities and how modalities affect each other Yan et al. (2025b). Open research questions are whether unifying these modalities could affect the final result, the training data recipe composition for different modalities, and how to *effectively mitigate modality hallucination, where the model over-relies on dominant signals (like text semantics) while ignoring subtle cues from others (like audio texture), to ensure balanced, high-fidelity generation across all streams* Wang et al. (2025g). Furthermore, designing unified tokenizers that can jointly compress video, text and audio into a shared latent space without suffering from interference or resolution loss remains a critical architectural hurdle Ma et al. (2025b).

## 7 Conclusion

This paper reviews the rapid shift from visual-only video synthesis toward native audio–visual generation, and provides the landscape around shared architectural building blocks, post-training strategies, evaluation protocols, and emerging application domains. We discuss how modern systems increasingly rely on scalable diffusion backbones and explicit cross-modal fusion/alignment to synchronize video dynamics with semantically and temporally consistent audio, while streaming, human-centric generation, and long-horizon storytelling/world-modeling are becoming the most active frontiers for real-world applications.

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
