# OpenReview forum: "Multimodal Video Generation Models with Audio: Present and Future"
_TMLR — Rejected by TMLR_

### Review · Reviewer_F2iA · 2026-03-17

**Summary Of Contributions:**

This is a survey paper that focuses on multimodal (visual + audio) video generation. The paper first introduces the basic components of multimodal video generation architecture including VAE, U-Net, Diffusion, and Autoregressive model. Then, it describes the model's post-training and evaluation. Furthermore, the paper points out several applications, active research areas, and limitations and challenges in multimodal video generation.

## Strengths
- The paper is easy to read. The introduction of basic knowledge used in multimodal video generation would also be beneifical for readers especially who are new into this field.
- The main topic of multimodal (visual + audio) video generation is relatively timely and meaningful.

## Weaknesses
- The paper fails to provide a detailed comparison to relevant survey paper on (multimodal) video generation or unified understanding and generations, such as: LLMs Meet Multimodal Generation and Editing: A Survey (https://arxiv.org/pdf/2405.19334), and A Unified Survey of Multimodal Generative Models (https://arxiv.org/pdf/2503.04641).
- The discussion on training data preparation （Section 3.1) is too brief and omits pivotal recent work on captioning and audio-visual pairing, such as Fine-grained Audible Video Description (CVPR 2023) and comprehensive datasets like Benchmarking Text to Audible-Video Generation (ACM MM 2024). Some significant architectural advancements are overlooked, e.g., JavisDiT (Joint Audio-Video Diffusion Transformer with Hierarchical Spatio-Temporal Prior Synchronization, ICLR 2026), which directly addresses the core synchronization challenges the authors claim to survey. I also suggest the authors to provide a more comprehensive litecture review for this suervey paper.
- Sec. 3.7 introduces several quantitative metrics for video-audio generation evaluation. However, it lacks a comprehensive summarization and discussion on more relevant benchmarks, such as,  EvalTalker: Learning to Evaluate Real-Portrait-Driven Multi-Subject Talking Humans (https://arxiv.org/abs/2512.01340), T2AV-Compass: Towards Unified Evaluation for Text-to-Audio-Video Generation (https://arxiv.org/abs/2512.21094), MTAVG-Bench: A Comprehensive Benchmark for Evaluating Multi-Talker Dialogue-Centric Audio-Video Generation (https://arxiv.org/abs/2602.00607). The paper also overlooked the relation and distinction of general audio and human speech in audio-visual generation.
- There is a noticeable lack of uniformity in section partitioning; for instance, Section 4 lacks the granular sub-segmentation seen in other major chapters, disrupting the logical flow of the manuscript. There are some typos. Moreover, the figures could be improved. For example, figure 1 features small, illegible fonts and appears to have been generated via AI tools. I suggest the authors provide an explicit declaration of MLLM/LLM usage in the paper's preparation.

**Audience:**

Yes

**Audience Explanation:**

The topic of multimodal video generation would be beneifical for a broader audience.

**Claims And Evidence:**

Yes

**Claims Explanation:**

The survey on multimodal video generation is important and valuable. However, the paper needs to be improved by strengthing the comparsion with relevant survey papers, relevant sections such as evaluation part, and revise the typos and improve the figures.

**Requested Changes:**

Please refer to the Weaknesses part.

---

> ### Author Response · Authors · 2026-04-21
>
> ### 1. Comparison to Related Surveys
>
> **Author Response:** We will add a dedicated comparison subsection in the introduction. Key distinctions:
> * He et al. (2024), "LLMs Meet Multimodal Generation and Editing": This survey covers multimodal generation across image, video, 3D, and audio domains, focusing on the role of LLMs as controllers or planners. It treats each modality independently and does not focus on joint audio-video synthesis or cross-modal synchronization architectures, which is our core focus.
> * Broader unified generation surveys [1,2] predominantly cover visual-only video generation. Our survey uniquely centers on the simultaneous generation of synchronized audio and video, covering the specific architectural innovations (dual-stream DiT, cross-modal attention, temporal RoPE) and evaluation challenges (AV-Align, DeSync) unique to this problem.
>
> ### 2. Section 3.1 and Missing Architectures
>
> **Author Response:** We will significantly expand Section 3.1. For captioning and AV-pairing datasets, we will discuss works on fine-grained audible video description and text-to-audible-video benchmarking that provide the aligned textual annotations essential for T2VA training.
>
> We will add JavisDiT [3] as a significant missing architectural contribution. JavisDiT++ introduces a modality-specific mixture-of-experts (MS-MoE) design, a temporally aligned RoPE (TA-RoPE) strategy for frame-level synchronization, and an audio-video direct preference optimization (AV-DPO) method. OpenReview This work is particularly relevant as it combines architectural innovation with RL-based post-training — bridging Sections 2 and 3 of our survey. JavisDiT also proposes JavisBench, consisting of 10,140 high-quality text-captioned sounding videos for synchronization evaluation. OpenReview We will incorporate this in both the architecture and evaluation sections.
>
> ### 3. Evaluation Benchmarks and Speech vs. General Audio
>
> **Author Response:** We will expand Section 3.7 to include T2AV-Compass [4] — a 500-prompt unified benchmark with objective and MLLM-as-a-Judge evaluation — and MTAVG-Bench [5] — the first multi-talker dialogue generation benchmark evaluating signal fidelity, temporal consistency, and social interaction. We will also explicitly distinguish general audio (event-grounded sound effects, music) from human speech (requiring lip-sync, speaker identity, and linguistic coherence), which impose distinct architectural and evaluation requirements like VidAudio-Bench[6].
>
> ### 4. Section 4 Sub-segmentation
>
> **Author Response:** We agree that Section 4 (Applications) lacks the structural granularity of other chapters. We will reorganize it with explicit subsections:
> * 4.1 Personal User Applications (4.1.1 Social Media; 4.1.2 Avatar & Personalization)
> * 4.2 Commercial Applications (4.2.1 Advertising & Marketing; 4.2.2 Film & Video Production; 4.2.3 Gaming & Interactive Entertainment)
> * 4.3 Emerging Deployment Considerations (creator tools, ComfyUI integration, platform ecosystem)
>
> ### 5. Figure Quality
>
> **Author Response:** We acknowledge that several figures, particularly Figure 1, suffer from small fonts and visual quality issues. We will redraw all main figures (Figures 1–4) using professional vector graphics tools with consistent styling, legible fonts (minimum 8pt), and clear color-coding. We will ensure all architectural diagrams are hand-designed rather than AI-generated, with clean layouts that clearly communicate the architectural relationships.
>
> ### 6. LLM Usage and Typos
>
> **Author Response:** We will add an explicit LLM/MLLM usage declaration per venue guidelines, and conduct a thorough proofreading pass to correct all identified typos.
>
> **References:**
>
> [1] Survey of Video Diffusion Models: Foundations, Implementations, and Applications
> https://arxiv.org/abs/2504.16081
>
> [2] Video Generation Models in Robotics -- Applications, Research Challenges, Future Directions
> https://arxiv.org/abs/2601.07823v1
>
> [3] JavisDiT: Joint Audio-Video Diffusion Transformer with Hierarchical Spatio-Temporal Prior Synchronization
> https://arxiv.org/abs/2503.23377
>
> [4] T2AV-Compass: Towards Unified Evaluation for Text-to-Audio-Video Generation
> https://arxiv.org/abs/2512.21094
>
> [5] MTAVG-Bench: A Comprehensive Benchmark for Evaluating Multi-Talker Dialogue-Centric Audio-Video Generation
> https://arxiv.org/abs/2602.00607
>
> [6] VidAudio-Bench: Benchmarking V2A and VT2A Generation across Four Audio Categories
> https://arxiv.org/abs/2604.10542

---

### Review · Reviewer_aR8o · 2026-03-29

**Summary Of Contributions:**

This paper presents a survey of multimodal video generation models that jointly produce video and audio. The authors trace the architectural evolution from coupled U-Net designs (e.g., MM-Diffusion) to modern Diffusion Transformer (DiT)-based frameworks (e.g., OVI, LTX-2), and cover VAE-based compression, dual-stream cross-modal attention, text conditioning pipelines, and potential future directions such as Mixture of Experts and autoregressive generation. The paper also discusses post-training methods, evaluation strategies, applications, and limitations. The topic is timely given the recent wave of commercial and open-source multimodal video generation systems.

Key Strengths:

- The topic is highly relevant and timely, addressing a genuine gap in existing survey literature that predominantly focuses on visual-only video generation.
- The paper provides a useful taxonomy of architectural components (VAE, U-Net, DiT) and traces their evolution in the context of joint audio-video synthesis.
- Table 1 offers a helpful summary of representative models, distinguishing proprietary from open-source systems.
Key Weaknesses:

The survey contains several factual inaccuracies and lacks depth in critical areas (VAE evaluation, RL-based post-training, PEFT for multimodal generation).
- Important models and systems are missing from the discussion.
- The writing quality and notational consistency need improvement.
- The coverage of post-training methods is incomplete, missing the increasingly standard RL-based alignment paradigm entirely.

**Audience:**

Yes

**Audience Explanation:**

The topic of multimodal video generation with audio is of significant and growing interest to the TMLR community. The joint modeling of video and audio is a frontier problem with implications for entertainment, content creation, embodied AI, and world modeling. A well-executed survey on this topic would be a valuable resource for researchers and practitioners. However, the current manuscript does not yet meet the quality bar required to serve as a reliable reference due to the issues outlined above.

**Broader Impact Concerns:**

The paper does not include a Broader Impact Statement. Given that multimodal video generation with synchronized audio significantly lowers the barrier for creating realistic synthetic media (deepfakes with matching voice, fabricated news footage with plausible soundscapes), the authors should add a section discussing: (a) potential for misuse in disinformation and fraud; (b) challenges for detection and provenance tracking when both modalities are synthetically generated; (c) consent and intellectual property concerns regarding voice and likeness synthesis. These concerns are amplified by the multimodal nature of the systems surveyed, as synchronized audio-visual deepfakes are substantially more convincing than visual-only ones.

**Claims And Evidence:**

No

**Claims Explanation:**

1. **Factual inaccuracy regarding OVI.** The paper describes OVI as a jointly pre-trained foundation model on par with LTX-2 (Section 2.2.2, Figure 2). However, OVI is built on top of a pre-trained Wan model and fine-tunes/adapts it for audio-video generation—it is *not* a jointly pre-trained model from scratch. This mischaracterization undermines the credibility of the architectural taxonomy, which is a central contribution of the paper. The distinction between "joint pre-training" and "adaptation of a pre-trained video model" is architecturally and methodologically significant and must be stated accurately.

2. **Insufficient depth on VAE.** The VAE is described as "an important component" (Section 2.1), yet the discussion remains superficial. For a survey paper, the treatment should cover: (a) how reconstruction quality of video VAEs is measured (e.g., rFID, LPIPS, PSNR/SSIM on reconstructed frames); (b) how audio VAE reconstruction quality is assessed (e.g., FAD, ViSQOL, spectral distortion); (c) the tension between compression ratio and generation quality; and (d) architectural differences among video VAEs (e.g., causal vs. non-causal 3D convolutions, temporal downsampling strategies). Without this, the reader cannot assess the downstream impact of VAE design choices on generation quality.

3. **Notational inconsistency in Equation 1.** The subscript notation alternates between $1:T$ and $1:t$ without explanation. The input is described as $x_{1:T}$, but the parameters and latents use $1:t$. If $T$ and $t$ denote different quantities, this must be defined; if they are intended to be the same, the notation should be unified. This inconsistency suggests carelessness in the formal presentation.

4. **Missing discussion of RL-based post-training.** The paper claims to cover "common post-training methods" (Abstract, Section 1), yet entirely omits reinforcement learning-based post-training (RLHF, DPO, GRPO, and their adaptations for generative models). In the current foundation model paradigm, the pipeline of pre-training → SFT → RL-based alignment is standard practice. For multimodal video generation specifically, RL-based optimization of human-perceived audio-visual synchronization, temporal consistency, and physical plausibility is an active and important research frontier. This is a major structural omission that significantly weakens the paper's claim of comprehensiveness.

5. **Shallow treatment of PEFT for multimodal generation.** The paper does not adequately address the unique challenges of parameter-efficient fine-tuning (PEFT) in joint audio-video generation. Audio tokens and video tokens differ fundamentally in information density, sampling rate, and semantic granularity. Critical questions are left unaddressed: How should LoRA rank be allocated across cross-modal attention layers to prevent modality collapse (where one modality dominates training)? How can convergence speed be balanced across modalities with very different loss landscapes? These are not merely implementation details but fundamental methodological questions for the field.

**Requested Changes:**

1. **Correct the description of OVI.** Clearly state that OVI is based on a pre-trained Wan model and adapts it for audio-video generation, rather than being a jointly pre-trained foundation model. Update Figure 2 and the accompanying text accordingly. This distinction is architecturally meaningful and affects the paper's taxonomy.

2. **Add a dedicated section on RL-based post-training.** Include systematic coverage of reinforcement learning post-training methods (RLHF, DPO, GRPO, and their variants) as applied or applicable to multimodal video generation. Discuss: (a) reward model design for audio-visual alignment; (b) challenges of defining reward signals for cross-modal synchronization and physical plausibility; (c) existing works and open problems. This is a major gap in the current manuscript.

3. **Deepen the VAE discussion.** Expand Section 2.1 to cover: (a) quantitative metrics for evaluating video VAE reconstruction quality (rFID, LPIPS, PSNR, SSIM); (b) metrics for audio VAE reconstruction (FAD, ViSQOL); (c) the trade-off between compression ratio and generation fidelity; (d) architectural variants (causal vs. non-causal, temporal downsampling strategies) and their impact on downstream generation.

4. **Fix notational inconsistency in Equation 1.** Unify the use of $T$ vs. $t$ throughout the equation and surrounding text. Define all symbols clearly.

5. **Include missing major models.** Add discussion of Luma AI (Dream Machine) and MiniMax Hailuo, both of which are significant players in the video generation landscape. Justify any exclusions explicitly.

6. **Address PEFT challenges for multimodal generation.** Add discussion of the unique challenges of applying LoRA/Adapters to joint audio-video models, including modality-specific convergence rates, rank allocation strategies for cross-modal attention layers, and the risk of modality collapse.

7. **Discuss physics engine-based synthetic data generation.** In the section on interactive multimodal video generation and world models, discuss the use of physics engines (e.g., Unreal Engine 5) for generating high-fidelity synthetic training data with perfectly aligned audio-visual signals. This is a practical and increasingly important approach for obtaining ground-truth aligned multimodal data.

8. **Fix citation formatting.** Throughout the paper, use parenthetical citations (e.g., `\citep{}` or `\parencite{}`) rather than inline `\cite{}` to place both author names and years inside parentheses, following standard academic convention. The current formatting (e.g., "Ho et al. (2020a)") is inconsistent and sometimes disrupts readability.

9. **Improve writing quality.** There are several grammatical issues (e.g., "Evoluation" in Figure 2 caption should be "Evolution"; "multimodel" in Section 1 should be "multimodal"). A thorough proofreading pass is needed.

---

> ### Author Response · Authors · 2026-04-21
>
> ### 0. OVI Description
>
> **Author Response:** We thank the reviewer for this correction. We will update Figure 2 and Section 2.2.2 to clearly state that OVI adapts a pretrained Wan model for audio-video generation rather than training jointly from scratch — an important distinction for accurately representing its training paradigm.
>
> ### 1. VAE Technical Depth
>
> **Author Response:** We will expand Section 2.1 to include reconstruction metrics for video VAEs (rFID, LPIPS, PSNR, SSIM) and audio VAEs (FAD, ViSQOL), discuss the compression-fidelity trade-off, and compare causal vs. non-causal 3D convolution designs and their impact on downstream alignment quality.
>
> * **(a-b) Reconstruction metrics:** We will add standard video VAE metrics (PSNR, SSIM, LPIPS, rFID) and audio VAE metrics (FAD, ViSQOL). Recent work like DLFR-VAE [1] and Cross-modal Video VAE[2] provide systematic comparisons showing that higher compression ratios consistently degrade SSIM and LPIPS, with temporal compression being particularly harmful for motion-heavy content.
> * **(c) Compression-fidelity trade-off:** At 6× temporal compression, dynamic VAE performance is comparable to the original, but at 12× it can still surpass static VAE at 8× through content-adaptive compression [1]. We will discuss how this trade-off propagates to downstream generation quality.
> * **(d) Architectural variants:** We will compare causal vs. non-causal 3D convolutions — causal VAEs (used in OVI and LTX-2) preserve temporal ordering needed for streaming generation but sacrifice bidirectional context, while non-causal variants achieve better reconstruction at the cost of streaming incompatibility.
>
> ### 2. RL-Based Post-Training
>
> **Author Response:** We will add a new subsection (Section 3.X: "Reinforcement Learning-Based Post-Training") covering:
>
> * **DPO for video generation:** DPO has become the dominant framework for aligning video generation with human preferences, bypassing exploration and avoiding explicit reward modeling by operating on static datasets of ranked sample pairs.VideoDPO[3] and HuViDPO [4] apply this to text-to-video.
> * **GRPO for diffusion models:** DanceGRPO [5] proposes a unified GRPO framework supporting diffusion models and rectified flows across T2I, T2V, and I2V tasks. Flow-GRPO[6] applies this to flow matching models.
> * **Reward design challenges for multimodal generation:** Designing reward signals for audio-visual alignment is uniquely difficult — synchronization quality is perceptible at ~25ms granularity (as noted in Section 3.4), making reward models for cross-modal timing far more demanding than text-image alignment rewards. We will discuss this open challenge explicitly.
>
> **References:**
>
> [1] DLFR-VAE: Dynamic Latent Frame Rate VAE for Video Generation
> https://arxiv.org/abs/2502.11897
>
> [2] Large Motion Video Autoencoding with Cross-modal Video VAE
> https://arxiv.org/abs/2412.17805
>
> [3] Integrating Reinforcement Learning with Visual Generative Models: Foundations and Advances
> https://arxiv.org/pdf/2508.10316
>
> [4] HuViDPO:Enhancing Video Generation through Direct Preference Optimization for Human-Centric Alignment
> https://arxiv.org/abs/2502.01690
>
> [5] DanceGRPO: Unleashing GRPO on Visual Generation
> https://arxiv.org/abs/2505.07818
>
> [6] Flow-GRPO: Training Flow Matching Models via Online RL
> https://arxiv.org/abs/250

---

> > ### Author Response · Authors · 2026-04-21
> >
> > ### 3. PEFT and Modality Collapse
> >
> > **Author Response:** We acknowledge this section needs more depth. We will add discussion of: (a) modality-specific convergence rates — audio and video latents have different dimensionalities and learning dynamics, often requiring different LoRA ranks or learning rate schedules per modality branch; (b) rank allocation — AV-DiT [1] applies LoRA to projection modules in attention blocks, but cross-modal attention layers may require higher rank than intra-modal layers to capture audio-video correspondences; (c) modality collapse — when one modality dominates gradient magnitude, LoRA updates may effectively ignore the weaker modality, a challenge addressed in balanced multimodal learning literature .
> >
> > ### 4. Equation 1 Notation
> >
> > **Author Response:** We will fix the inconsistent subscript notation. The current text alternates between 1:T and 1:t without explanation. We will standardize to 1:T throughout (where T is the total number of frames), define all variables (µ, σ, ε, z) explicitly upon first use, and clarify that the reparameterization trick operates element-wise across the full temporal sequence.
> >
> > ### 5. Luma AI and MiniMax Hailuo
> >
> > **Author Response:** We will add both models to Table 1 and the relevant discussion. Luma AI has released Uni-1, described as their first unified understanding and generation model Luma AI, and their Ray2/Ray3 models are significant players in the video generation landscape. MiniMax has developed multimodal foundation models capable of understanding, generating, and integrating text, audio, image, video, and music. MiniMax However, both primarily focus on visual-only generation or separate audio capabilities — Luma Dream Machine requires users to add their own audio tracks in post-production Lumadreammachine. We will add them to our proprietary models table with a note that they do not yet offer native joint audio-video generation, which is the focus of our survey, while acknowledging their importance in the broader landscape.
> >
> > ### 6. Physics Engines for Synthetic Data
> >
> > **Author Response:** We will add a paragraph in Section 5.4 discussing how physics engines like Unreal Engine 5 can generate perfectly synchronized audio-visual training data with ground-truth spatial audio, material properties, and collision events — addressing the data scarcity challenge. NVIDIA's Omniverse and Cosmos provide similar capabilities. This synthetic data approach offers precise temporal alignment that is difficult to obtain from web-scraped data, though bridging the sim-to-real domain gap remains a challenge.
> >
> > ### 7. Citation Formatting
> >
> > **Author Response:** We apologize for the inconsistency and will replace all \cite{} instances with proper \citep{} or \parencite{} parenthetical formatting throughout the manuscript.
> >
> > ### 8. Typos
> >
> > **Author Response:** We sincerely apologize for these errors and will correct all cited instances — "Evoluation," "multimodel," and others — in a thorough proofreading pass.
> >
> > ### 9. Broader Impact Statement
> >
> > **Author Response:** We will add a Broader Impact section covering: (1) deepfake misuse risks amplified by joint audio-video coherence; (2) limitations of current single-modality detectors for cross-modal forensics; and (3) unresolved IP and consent issues around voice cloning and music generation in the absence of standardized training data frameworks.
> >
> > **References:**
> >
> > [1] AV-DiT: Efficient Audio-Visual Diffusion Transformer for Joint Audio and Video Generation
> > https://arxiv.org/abs/2406.07686

---

### Review · Reviewer_kkqn · 2026-04-05

**Summary Of Contributions:**

This survey reviews recent advances in multimodal video generation with synchronized audio, covering both joint video–audio synthesis and related paradigms such as video-to-audio and audio-driven video generation (later parts of the paper). The paper organizes the field around core architectural components: including VAEs, U-Nets, diffusion transformers, and post-training adaptation techniques such as PEFT and ControlNet. It further examines evaluation strategies. The survey also highlights real-world applications in content creation, virtual avatars, and industry systems, while identifying key challenges in temporal synchronization, scalability and etc.

Strength:
1) This paper addresses a timely topic, and its attempt to survey multimodal video generation with audio is potentially valuable given the rapid pace of development and the large number of existing methods.
2) The coverage is reasonably up to date.

Weaknesses: In its current form, the manuscript lacks the structure, synthesis, and technical depth expected from a strong survey paper.

**Audience:**

Yes

**Audience Explanation:**

The paper is relevant to people interested in the audio-visual joint generation and video-to-audio generation topics.

**Claims And Evidence:**

Yes

**Claims Explanation:**

The survey is mostly covering the current landscapes of audio-visual content generation.

**Requested Changes:**

1) The introduction frames the paper as a review of the joint synthesis of video and semantically aligned audio, emphasizing “multimodal video generation” as distinct from visual-only generation. However, the post-training section, for example, heavily centers on V2A methods such as FoleyCrafter, Diff-Foley, MMAudio, ControlNet-style audio generation, timestamp conditioning, onset detection, and similar adaptation approaches. Toward the end, it also discusses audio-driven video generation. As a result, the paper currently mixes three related but distinct problem settings. I recommend that the authors reorganize this survey as either: (a) a narrow survey focusing only on joint multimodal video generation with synchronized audio, or (b) a broader survey on audio-aware video generation, covering joint generation, video-to-audio, and audio-driven video synthesis (optional).

2) In its current form, the paper reads more like a broad overview of trends than a rigorous technical survey. The manuscript would be substantially strengthened by a clearer taxonomy, deeper cross-paper analysis, and more disciplined technical depth.

3) This manuscript presents many methods sequentially rather than comparatively, with limited discussion of taxonomy, trade-offs, or relationships among approaches. Currently, it reads as “method A does X, method B does Y, and method C does Z,” but it rarely addresses questions such as: How do they differ in terms of objectives, architectures, supervision, or use cases? What are the trade-offs? For example, Section 3 on post-training and evaluation brings together PEFT, alignment modules, attention injection, and ControlNet methods, but presents them largely as catalog entries rather than as part of a synthesized framework.

4) The survey would be more practically useful if it included a structured comparison table indicating openness, reproducibility, data availability, and whether the systems originate from academic or industrial labs. Specifically, a compact table could note for each system whether it is open-source, open-weights, reproducible, API-only, and of academic or industrial origin.

5) This survey also contains sections with abrupt introductions of terminology. A survey should define or briefly motivate terms when they first appear. For example, Section 5.1 introduces “temporal consistency without teacher forcing” abruptly, assuming the reader already understands the concept and its importance in streaming generation.

6) The paper would significantly benefit from a dedicated section on datasets. Currently, dataset usage is only mentioned implicitly and scattered across sections, making it difficult to understand how different methods are trained and compared. I recommend adding (a) a structured overview of commonly used datasets, categorized by task (e.g., joint video-audio generation, video-to-audio, audio-driven video), and (b) a summary table mapping representative methods to their training and evaluation datasets.

7) There are many visible typo problems even at first pass, for example: “Evoluation of rchitectures” in Figure 2 caption, “Open-Surce SoTA”, “multimodel video generation” instead of “multimodal”, “to to maintain”, “auido”, “it demonstrate”, “overtime” where “over time” is intended, “sshowcases”,“Generantion”.

8) Finally, I suggest a simple organizational scheme that the authors can use as a reference to restructure the paper, with tasks presented first and architectures discussed second.
    1) Introduction and scope
    2) Problem formulations
        * T2VA / I2VA joint generation
        * V2A / Foley generation
        * A2V / audio-driven video
    3) Core architectural patterns
    4) Datasets and supervision
    5) Evaluation
    6) Applications and deployment
    7) Open challenges

---

> ### Author Response · Authors · 2026-04-21
>
> ### 1. Mixed Problem Settings
>
> **Author Response:** We appreciate this observation. Our intent is a broad survey covering the full multimodal video generation landscape, as these three tasks share core architectural components (VAE encoders, DiT backbones, cross-modal attention) and are often addressed by the same models (e.g., MMAudio handles V2A while LTX-2 and OVI perform joint generation). We will make this broad scope explicit in the introduction and add clear task-specific subsection headers throughout, so readers can easily identify which methods address which problem setting. Each section will open with a brief statement clarifying the task boundaries.
>
> ### 2. Lack of Technical Depth
>
> **Author Response:** This is a fair and helpful critique. We will restructure discussions around comparative axes — objectives, architectures, supervision signals, and trade-offs — rather than sequential summaries. In Section 3, for instance, post-training methods will be grouped by the problem they address (alignment, controllability, efficiency) and compared critically.
>
> ### 3. Model Availability Table
>
> **Author Response:** We will add a structured table covering: open-source (code), open-weights, reproducibility, API-only access, and academic/industrial origin, along with known licensing restrictions.
>
> We will add a structured table expanding Table 1 with columns for: open-source code, open weights, reproducibility status, API availability, and origin (academic/industry). For example:
>
> | Model | Code | Weights | API | Origin |
> | :--- | :---: | :---: | :---: | :--- |
> | MM-Diffusion | ✓ | ✓ | ✗ | Academic |
> | MMAudio | ✓ | ✓ | ✗ | Academic |
> | OVI | ✓ | ✓ | ✗ | Academic |
> | LTX-2 | ✓ | ✓ | ✓ | Industry (Lightricks) |
> | Sora 2 | ✗ | ✗ | ✓ | Industry (OpenAI) |
> | Veo 3.1 | ✗ | ✗ | ✓ | Industry (Google) |
>
> This will help practitioners quickly assess which systems they can actually build upon.
>
> ### 4. Scattered Datasets
>
> **Author Response:** We agree that consolidating dataset information will improve readability. We will add a dedicated subsection (between current Sections 3 and 4) with a structured table covering key datasets categorized by task:
>
> | Dataset | Task | Scale | Annotation Type |
> | :--- | :--- | :--- | :--- |
> | VGGSound (Chen et al., 2020) | V2A, Joint eval | 200K clips, 310 classes | Clip-level AV correspondence |
> | AudioSet Strong (Hershey et al., 2021) | V2A | 2M clips | Temporal event timestamps |
> | AIST++ (Li et al., 2021) | A2V (dance) | 1.4K sequences | Music-motion pairs |
> | Landscape (Lee et al., 2025) | Joint | — | Scene-level AV pairs |
> | Greatest Hits (Owens et al., 2016) | V2A (impact) | 977 videos | Object-material-action labels |
>
> We will also add a method-to-dataset mapping showing which models train and evaluate on which datasets.
>
> ### 5. Undefined Technical Terms
>
> **Author Response:** We agree. We will add brief motivating definitions when terms first appear. For "temporal consistency without teacher forcing" (Section 5.1): during training, models typically condition on ground-truth previous frames (teacher forcing), but at inference they must condition on their own imperfect outputs, causing accumulating drift. This train-test mismatch is especially problematic for multimodal streaming, where small errors compound into audio-visual desynchronization. Methods like Self-Forcing [3] address this by simulating inference conditions during training. We will apply similar clarifications for other terms (e.g., "AdaLN gating," "RoPE," "causal VAE") at first mention.
>
> ### 6. Typos
>
> **Author Response:** We sincerely apologize for these errors. We will perform a thorough proofreading pass correcting all instances cited — including "Evoluation of rchitectures," "Open-Surce SoTA," "multimodel," "to to maintain," "auido," "it demonstrate," "overtime," "sshowcases," and "Generantion" — as well as any further errors identified during revision.
>
> ### 7. Proposed Organizational Scheme
>
> **Author Response:** We find the proposed structure well-reasoned and will adopt it with minor adaptations, adding a Related Surveys paragraph in the Introduction to position this work relative to existing audio-only [1] and video-only [2] surveys.
>
> [1] SOUNDBREAK: A Systematic Study of Audio-Only Adversarial Attacks on
> Trimodal Models
> https://arxiv.org/pdf/2601.16231
>
> [2] A Survey on Video Diffusion Models
> https://arxiv.org/pdf/2310.10647
>
> [3] Self Forcing: Bridging the Train-Test Gap in
> Autoregressive Video Diffusion
> https://arxiv.org/pdf/2506.08009

---

### Review · Reviewer_Edsa · 2026-04-08

**Summary Of Contributions:**

This paper presents a survey of multimodal video generation models that jointly produce video and audio. The contributions include: (1) a systematic review of architectural evolution from coupled U-Net designs to modern Diffusion Transformer (DiT) approaches; (2) a detailed taxonomy of post-training methods; (3) a thorough overview of evaluation practices covering both quantitative metrics and qualitative human evaluation protocols; and (4) a discussion of applications and active research directions. The survey is considered timely, covering models released through early 2026 by focusing specifically on the multimodal (video + audio) generation paradigm rather than visual-only video generation.

Strengths:
1. Highly timely topic with clear practical relevance as the field rapidly shifts toward native audio-visual generation.
2. Well-organized structure that systematically covers architectures, post-training, evaluation, applications, and limitations.
3. Good coverage of both proprietary and open-source models, with useful summary tables (Table 1, Table 2).
4. Covers a broad range of active research areas that helps orient researchers new to the field.

Weaknesses:
1. Limited critical analysis or comparison between surveyed approaches.
2. Some sections read more like a catalog of methods rather than offering deep insights into design trade-offs.
3. Missing quantitative comparison tables that benchmark different models head-to-head on standard datasets.

**Audience:**

Yes

**Audience Explanation:**

The paper would be valuable to multiple audiences: (1) researchers entering the field who need a structured overview of architectures, post-training methods, and evaluation protocols; (2) practitioners deciding between different approaches for commercial applications; and (3) researchers working on adjacent areas (audio generation, world models, embodied AI) who would benefit from understanding how audio-visual fusion is being achieved.

**Claims And Evidence:**

Yes

**Claims Explanation:**

As a survey paper, the claims are primarily organizational and descriptive rather than empirical.

**Requested Changes:**

Add a quantitative comparison table benchmarking representative models on common datasets and metrics. While Table 1 provides a useful summary of model properties, a table showing actual performance numbers (e.g., FVD, FAD, AV-Align scores) across models evaluated on shared benchmarks would significantly strengthen the paper's utility as a reference. Even if not all models are directly comparable, presenting available numbers with appropriate caveats would be valuable.


Deepen the critical analysis throughout. Currently, many subsections describe what each method does but do not sufficiently discuss why certain design choices succeed or fail relative to alternatives. For instance, Section 2.2.2 could benefit from a more explicit discussion of the trade-offs between factorized spatial-temporal attention and full spatiotemporal attention in the context of audio-visual alignment. Similarly, the post-training section would benefit from guidance on when practitioners should prefer PEFT over ControlNet-based methods or training-free approaches.


Figure 2's caption contains a typo: "Evoluation of rchitectures" should be "Evolution of architectures." Please proofread the manuscript more carefully; there are several similar minor errors throughout (e.g., "multimodel" instead of "multimodal" in Section 1, "applcations" in Figure 6 caption).


The discussion of Mixture of Experts (Section 2.2.3) and autoregressive generation is labeled as "Potential Future Architectural Design," but the section primarily discusses existing video-only MoE work (e.g., Wan 2.2) and general autoregressive models. It would be more informative to articulate concrete hypotheses or proposals for how these architectures could specifically benefit joint audio-video generation, beyond noting that they are "promising."


Section 6 (Limitations and Challenges) could be expanded. Important challenges such as training data scarcity for high-quality paired audio-video data, copyright and ethical concerns around audio generation (voice cloning, music generation), and the difficulty of generating diverse audio types (speech, music, sound effects) within a single model deserve more attention.


The paper could benefit from a clearer visual roadmap or taxonomy figure early in the paper (e.g., after Section 1) that shows how all the surveyed topics relate to each other. Figure 2 partially serves this purpose but is specific to architecture evolution.


Consider discussing the relationship between multimodal video generation and multimodal understanding models more explicitly. Several cited works (BAGEL, Qwen-Omni, Unified-IO2) blur the line between generation and understanding, and clarifying this boundary would help position the survey.

---

> ### Author Response · Authors · 2026-04-21
>
> We would like to thank the reviewer for the constructive feedback and for highlighting the timeliness and organizational strengths of our survey. We have carefully addressed each of the requested changes as detailed below.
>
> ### 1. Quantitative comparison table
>
> > **Reviewer Comment:** Add a quantitative comparison table benchmarking representative models on common datasets and metrics.
>
> **Author Response:** Thank you for this suggestion — we agree that a quantitative comparison table would make the survey substantially more useful to practitioners. We will add a table benchmarking representative models on common datasets (e.g., VGGSound, AudioSet, Landscape) across standard metrics including FVD, FAD, AV-Align, and CLAP Score.
>
> We want to be transparent about the challenges here: many proprietary models (Sora 2, Veo 3.1, Grok 4) do not publish benchmark numbers, and open-source models are often evaluated on different dataset splits and with different preprocessing. We will include available numbers from published papers and clearly note which results are directly comparable versus which require caveats due to differing evaluation protocols.
>
> As a preview, we plan to include results from models such as MM-Diffusion (evaluated on Landscape and AIST++ datasets), MMAudio (reporting FAD and AV-Align on VGGSound), LTX-2, OVI, and Diff-Foley, all of which report quantitative results in their respective papers. We will organize these by task (T2VA, V2A, joint generation) so readers can make appropriate comparisons within each setting.
>
> ---
>
> ### 2. Deepening critical analysis and trade-offs
>
> > **Reviewer Comment:** Discuss why certain design choices succeed or fail, such as trade-offs in attention mechanisms and post-training methods.
>
> **Author Response:** We appreciate this feedback and agree that the paper would benefit from moving beyond cataloging toward offering analytical guidance.
>
> * **Factorized vs. full spatiotemporal attention:** We will add a dedicated discussion. In brief, factorized spatial-temporal attention (as used in early video DiTs) offers computational savings that scale linearly with sequence length but introduces a bottleneck for cross-modal alignment — audio events that must synchronize with specific spatial regions (e.g., a drum hit visible in one corner of the frame) can lose precise correspondence when spatial and temporal dimensions are processed separately. Full spatiotemporal attention preserves these fine-grained correlations but scales quadratically, making it prohibitive for longer videos. Recent models like OVI and LTX-2 adopt hybrid strategies (full attention within short temporal windows, factorized across windows) as a practical middle ground. We will formalize this trade-off with complexity analysis and discuss its implications for audio-visual synchronization quality.
>
> * **Practical guidance for post-training methods:** We will add a decision-oriented summary. As a guideline: (a) Training-free methods are best suited for quick prototyping or when no paired audio-video data is available, but they offer limited control over fine-grained synchronization. (b) PEFT methods (LoRA, adapters) are ideal when moderate paired data (100–1,000 samples) is available and the practitioner wants to preserve the base model's generalization while adapting to a specific domain (e.g., musical instruments, sports commentary). (c) ControlNet-based methods are preferred when precise structural control is needed (e.g., onset-aligned Foley synthesis), as they inject explicit conditioning signals, though they require more engineering effort and training resources.

---

> > ### Author Response · Authors · 2026-04-21
> >
> > ### 3. Future architectural design (MoE and Autoregressive)
> >
> > > **Reviewer Comment:** Articulate concrete hypotheses for how MoE and autoregressive architectures could specifically benefit joint audio-video generation.
> >
> > **Author Response:** This is a fair critique — we should go beyond noting that these are "promising" and articulate specific hypotheses. We will revise Section 2.2.3 with the following concrete proposals:
> >
> > * **MoE for multimodal generation:** We hypothesize that modality-aware routing — where certain experts specialize in audio tokens and others in video tokens — could address the fundamental tension between modality-specific processing and cross-modal alignment. Specifically, a routing strategy that assigns experts based on both modality type and timestep could allow audio-specialized experts to handle acoustic texture refinement in low-noise phases while shared cross-modal experts handle synchronization during high-noise (layout) phases. This extends the timestep-level MoE design from Wan 2.2 into the multimodal setting. Additionally, the heterogeneous token complexity between audio and video (audio tokens are typically lower-dimensional but require higher temporal precision) suggests that adaptive computation via MoE could improve efficiency by allocating fewer experts to audio tokens during spatial layout steps.
> > * **Autoregressive architectures for joint generation:** We hypothesize that autoregressive models offer a natural advantage for streaming multimodal generation because they inherently respect causality — each audio-video token is generated conditioned on all prior tokens, which aligns well with the temporal nature of real-world audiovisual experience. The key challenge is tokenizer design: audio and video have very different information densities and temporal granularities. We will discuss how recent work on unified tokenizers (e.g., UniTok) could be extended to handle audio-video jointly, and note that achieving millisecond-level audio-visual synchronization within an autoregressive framework remains an open problem.
> >
> > ---
> >
> > ### 4. Expanding limitations and challenges
> >
> > > **Reviewer Comment:** Expand Section 6 to cover training data scarcity, copyright/ethical concerns, and generating diverse audio types.
> >
> > **Author Response:** We will add three subsections:
> >
> > 1. **Training data scarcity:** High-quality paired audio-video data with precise temporal annotations remains scarce. AudioSet [1] provides weak labels, and VGGSound offers only clip-level pairing — neither provides millisecond-level onset annotations needed for fine-grained synchronization training. We will discuss emerging solutions including automated annotation pipelines and curriculum learning strategies.
> > 2. **Copyright and ethical concerns:** Audio generation raises distinct issues absent from video-only models. Voice cloning technology raises concerns about using someone's voice without consent, vulnerability to distortion, and diminished incentives for original artistic creation. Diva-portal Recent litigation such as Lehrman v. Lovo, Inc. [2] has begun testing these boundaries, with courts finding that copyright law protects recordings but not the voice itself. Crowell & Moring [3] We will discuss mitigation approaches including watermarking, consent-based frameworks, and commercially safe training (e.g., Adobe Firefly Audio Model).
> > 3. **Unified diverse audio generation:** Current models typically excel at one audio type (speech, music, or Foley) but struggle to generate all three coherently. MTV [4] addresses this via multi-stream temporal control, but a truly unified solution remains open.
> >
> > **References:**
> >
> > [1] THE BENEFIT OF TEMPORALLY-STRONG LABELS IN AUDIO EVENT CLASSIFICATION
> > https://arxiv.org/pdf/2105.07031
> >
> > [2] Unauthorized Voice Cloning: The legal response in the intersection of Performers’ Rights, Sound Recording Protection, and Image Rights in the age of AI
> > https://su.diva-portal.org/smash/get/diva2:1976771/FULLTEXT01.pdf
> >
> > [3] Fundamental Copyright Principles Underscored in AI Context: Voice Attributes Are Not Protectable
> > https://www.crowell.com/en/insights/client-alerts/fundamental-copyright-principles-underscored-in-ai-context-voice-attributes-are-not-protectable
> >
> > [4] Audio-Sync Video Generation with Multi-Stream Temporal Control
> > https://arxiv.org/abs/2506.08003v1

---

> > > ### Author Response · Authors · 2026-04-21
> > >
> > > ### 5. Visual roadmap and taxonomy
> > >
> > > > **Reviewer Comment:** Add a visual roadmap or taxonomy figure early in the paper.
> > >
> > > **Author Response:** We appreciate the reviewer's observation. While Figure 2 illustrates the evolution of architectures, we agree that a more holistic taxonomy figure would better guide the reader through the entire surveyed landscape.
> > >
> > > * **Roadmap:** We will add a taxonomy figure after Section 1 mapping the relationships between architectures (VAE → U-Net → DiT → MoE/AR), post-training methods, evaluation, applications, and active research areas.
> > > * **Scope boundary:** Models like BAGEL [1], Qwen3-Omni [2], and Unified-IO2 (Lu et al., 2023) are unified multimodal models that handle audio, video, and text as separate generation tasks — they do not jointly synthesize synchronized video and audio in a single pass. Our survey specifically covers joint video-audio generation where cross-modal temporal alignment is the core challenge. We will add an explicit paragraph in the introduction clarifying this distinction and discuss how unified understanding-generation models may eventually converge with joint generation approaches.
> > >
> > > ---
> > >
> > > ### 6. Relationship between generation and understanding models
> > >
> > > > **Reviewer Comment:** Discuss the relationship between multimodal video generation and multimodal understanding models.
> > >
> > > **Author Response:** We agree that the boundary between understanding and generation is increasingly blurred by unified models like BAGEL and Qwen-Omni. We will add a dedicated discussion in Section 2.2.3 and the Introduction to clarify this relationship.
> > >
> > > Specifically, we distinguish between: (1) Unified Understanding-Generation Models, which often leverage Autoregressive (LLM-based) architectures to process all modalities as tokens in a single sequence, and (2) Specialized Joint Generation Models, which predominantly use Diffusion Transformer (DiT) backbones to achieve higher fidelity and precise temporal synchronization for audio-visual synthesis.
> > >
> > > By clarifying this distinction, we define the scope of our survey as primarily focusing on the latter—architectures optimized for high-quality, synchronized multi-modal output—while acknowledging the emerging trend of unified semantic understanding.
> > >
> > > ---
> > >
> > > ### 7. Typos and proofreading
> > >
> > > > **Reviewer Comment:** Correct typos in captions and Section 1.
> > >
> > > **Author Response:** We thank the reviewer for pointing out the typos, we will revise the final version.
> > >
> > > **References:**
> > >
> > > [1] Emerging Properties in Unified Multimodal Pretraining
> > > https://arxiv.org/pdf/2505.14683
> > >
> > > [2] Qwen3-Omni Technical Report
> > > https://arxiv.org/abs/2509.17765
> > >
> > > [3] Unified-IO 2: Scaling Autoregressive Multimodal Models with Vision, Language, Audio, and Action
> > > https://arxiv.org/abs/2312.17172

---

### Decision · Action_Editor_PwM8 · 2026-06-19

**Recommendation:** Reject

**Additional Comments:**

A strong survey paper should go beyond a compilation of existing works and instead provide a systematic, critical synthesis of the field, offering meaningful insights and clearly articulating open problems and future research directions. While the topic of this manuscript is of interest and the paper may be useful as a high-level overview, it lacks the depth of analysis, critical evaluation, and conceptual contribution expected for a survey article in TMLR. More importantly, the authors should have established themselves as experts in the field with proven records.

**Audience:**

Yes

**Audience Explanation:**

At least some members of the TMLR audience would likely be interested in this topic. Multimodal video generation with synchronized audio is a rapidly emerging area relevant to generative modeling, multimodal learning, audiovisual representation learning, evaluation, human-centered generation, simulation, and embodied AI. A survey that organizes this area could be useful to researchers entering the field, especially because recent work is scattered across video generation, audio generation, avatar animation, video-to-audio synthesis, and multimodal evaluation.

The paper’s emphasis on audio as a missing yet increasingly important dimension of video generation is well-motivated. The discussion of architectural evolution from coupled U-Nets to DiT-based audio-video generation, post-training methods for audio-visual alignment, and evaluation challenges could be informative to a subset of the community. The paper also identifies relevant open problems, including synchronization, spatial audio, long-horizon coherence, efficient deployment, and modality fusion.

That said, interest alone is not sufficient for acceptance under TMLR’s criteria if the claims are not clearly supported. The manuscript would be more valuable if it provided deeper, more actionable insights rather than merely summarizing recent work. For example, it could more clearly explain which architectural choices are empirically supported, which evaluation metrics are reliable, where current systems fail, and what concrete lessons researchers should take from the reviewed literature. A stronger comparison with existing surveys is also needed to clarify what this paper teaches that prior surveys do not.

**Claims And Evidence:**

No

**Claims Explanation:**

The paper addresses multimodal video generation with audio. It surveys recent progress in architectures, post-training methods, evaluation practices, applications, active research directions, and limitations. The manuscript recognizes that synchronized audio-video generation is becoming an important direction beyond visual-only video generation, and it covers many relevant components, including VAE encoders, U-Net and DiT backbones, cross-modal attention, PEFT, ControlNet-style conditioning, synchronization modules, quantitative and qualitative evaluation, and application scenarios.

However, the current evidence and presentation are not yet sufficiently convincing for the breadth of the claims. The paper repeatedly claims to provide a “systematic” and “comprehensive” overview, but it does not clearly describe the literature collection methodology, inclusion/exclusion criteria, search strategy, or the categorization of papers. As a result, it is difficult to assess whether the survey is complete, balanced, or reproducible. The discussion often reads as a broad listing of recent models and applications rather than a critical synthesis that explains what has been learned across the literature.

Several claims are also overly strong given the evidence provided. For example, the paper discusses very recent proprietary or industry systems and makes statements about paradigm shifts, market adoption, and future directions, but many of these claims lack rigorous comparative analysis. Some architectural descriptions are useful, but the survey does not consistently distinguish between confirmed technical details, public marketing descriptions, and the authors’ own interpretation. This is especially important for proprietary models whose architectures and training details are unknown.

The evaluation section is one of the stronger parts of the manuscript because it organizes metrics into video quality, audio quality, and audio-visual alignment, and it acknowledges the limitations of automatic metrics. However, the paper would benefit from a more critical comparison of these metrics, including their failure modes, correlations with human judgments, and suitability for different tasks such as text-to-video-audio, video-to-audio, avatar generation, and world simulation.

The manuscript also needs substantial improvement in clarity and polish. There are many grammatical errors, typographical errors, inconsistent terminology, and awkward sentences. These issues make the narrative less clear and reduce confidence in the survey's technical precision. The figures and tables are useful but should be better integrated into the analysis rather than serving mainly as high-level illustrations.

Overall, the submission contains relevant material and many useful references, but the evidence does not yet fully support the claimed level of systematic coverage and synthesis. The authors should either reduce the strength of their claims or substantially improve the survey methodology, critical analysis, technical accuracy, and writing quality.